# Explicit stochastic advection algorithms for the regional scale particle-resolved atmospheric aerosol model WRF-PartMC (v1.0)

Jeffrey H. Curtis[1,2], Nicole Riemer[1], and Matthew West[2]

[1]Department of Climate, Meteorology & Atmospheric Sciences, 1301 W. Green St, Urbana, IL 61801, USA
[2]Department of Mechanical Science and Engineering, University of Illinois Urbana-Champaign, 1206 W. Green St., Urbana, IL 61801, USA

**Correspondence:** Jeffrey H Curtis (jcurtis2@illinois.edu)

**Abstract.** This paper presents the development of a stochastic particle method to simulate advection in regional-scale models with a particle-resolving aerosol representation. The new method is based on finite volume discretizations with the flux terms interpreted as probabilities of particle transport between grid cells. We analyze the method in 1D and show that the stochastic particle sampling during transport injects energy at high spatial frequencies, which can be partially compensated for with the choice of a dissipative odd-order finite volume scheme. We then apply the stochastic third- and fifth-order advection algorithms with monotonic limiters in WRF-PartMC, using both idealized and realistic wind fields in 2D and 3D. In all cases we observe the expected convergence rates of the stochastic particle method to the finite volume solution as the number of computational particles is increased. This work enables the use of particle-based aerosol models on the regional scale.

## 1 Introduction

Aerosol particles influence the climate system as cloud condensation nuclei (CCN), as ice nucleating particles, and as scatterers and absorbers of radiation (Masson-Delmotte et al., 2021). Estimating the magnitude of the aerosol impact on climate requires not only the information of bulk aerosol composition and size distribution, but also the information of aerosol mixing state (Riemer et al., 2019), i.e., the way the chemical species are distributed across the particle population (Winkler, 1973). The aerosol mixing state can vary between a fully external mixture, where each particle contains only one chemical species which can differ between different particles, and a fully internal mixture where each particle is composed of the same mixture of species. In reality, the mixing state is in between these two extreme cases (Bondy et al., 2018; O'Brien et al., 2015; Ye et al., 2018; Healy et al., 2014). Furthermore, many physical and chemical processes change the mixing state during the aerosol's lifetime in the atmosphere (Li et al., 2016). Representing these processes in models poses large challenges but is needed to predict the aerosol climate impact (Bauer et al., 2013; Fierce et al., 2017).

Atmospheric three-dimensional chemical transport models or Earth system models utilize a variety of aerosol representations that differ in their levels of detail. These can be categorized into bulk approaches (Koch, 2001; Tegen and Miller, 1998), modal modeling approaches (Whitby and McMurry, 1997), and sectional modeling approaches (Seigneur et al., 1986). These methods have in common that they do not fully resolve the mixing state of the aerosol, but instead use a priori assumptions. For example, modal models assume that each mode is internally mixed, while different modes can differ in the set of species

that they track. Sectional models capture the size dependence of aerosol composition but within one section only the average aerosol composition is known. These approaches can be refined by introducing additional modes (Bauer et al., 2008; Liu et al., 2012, 2016), additional one-dimensional sectional distributions (Jacobson, 2002; Zhang et al., 2014) or additional dimensions to the bin structure itself (Matsui et al., 2013; Matsui, 2016; Zhu et al., 2015; Ching et al., 2016), where each dimension represents one species or group of species. Comparing these more sophisticated types of models against versions that use

more simplified mixing state representations shows that mixing state approximations impact the estimation of both optical and CCN properties and contribute to the structural and parametric model uncertainties. For example, Zhu et al. (2016) performed simulations with a sophisticated mixing-state-aware model (SCRAMS) for the region of Paris, France. Different mixing state treatments caused differences in aerosol water uptake, which propagated into differences in aerosol optical depths of up to 70%. Lee et al. (2016) carried out simulations with a mixing-state-resolving (source-oriented) version of WRF-Chem for the

region of the Californian Central Valley. They found a decrease in the ratio of CCN to total aerosol number concentration from 94% with an internal mixture assumption to 80% with a more detailed source-oriented mixture. Furthermore, the range of uncertainties can depend on the degree to which mixing state is represented. This was shown by Matsui et al. (2018) who quantified the sensitivity of the present-day BC direct radiative effect due to uncertainties in emission size distributions. They found that the uncertainty is 5–7 times larger when the BC mixing state is sufficiently resolved compared to a simplified model

representation where an internal mixture is assumed.

It is important to note that the storage requirements for multi-dimensional bin structures grow exponentially with the number of species (the curse of dimensionality). Therefore, in practice, the multi-dimensional bin approach is limited to two or three dimensions, whereas the composition space of the atmospheric aerosol contains tens or even hundreds of species. Hence, although this model approach carries more detail than 1D bin structures, it is still not able to resolve the mixing state fully.

In contrast to the above mentioned distribution-based methods, particle-resolved methods provide a different approach to representing the atmospheric aerosol (Riemer et al., 2009; Shima et al., 2009; Grabowski et al., 2019). They use a collection of discrete computational particles, where each particle can be thought of as a vector that stores the masses of each aerosol species and other particle attributes (e.g., information about particle shape or particle source) and that evolves over the course of the simulation. Aerosol mixing state is therefore intrinsically resolved and does not require any ad hoc assumptions. Furthermore,

it is straightforward to add more attributes to the particles as this does not result in an exponential increase of storage. Instead, it scales linearly with the number of particles. Particle methods are therefore beneficial for problems where high-dimensional data is involved as they break the curse of dimensionality.

In this paper we describe the development of stochastic advection algorithms that enable the particle-resolved aerosol model PartMC to be used on the regional scale, embedded within the Weather Research and Forecast (WRF) model. While we

only present the development of stochastic advection schemes based on the finite volume methods in WRF, the methodology described here is applicable to any finite volume scheme or transport scheme such as Corner-Transport Upwind (Colella, 1990; LeVeque, 2002) or Flux-Form Semi-Lagrangian (Lin and Rood, 1996, 1997) that can be found in other host models. This paper builds on previous work of developing the stochastic, particle-resolved PartMC-MOSAIC box model (Riemer et al., 2009; DeVille et al., 2011; Curtis et al., 2016; DeVille et al., 2019), and the one dimensional single-column model WRF-

PartMC-MOSAIC-SCM (Curtis et al., 2017). These modeling tools have been used to investigate the black carbon aging process (Riemer et al., 2010; Fierce et al., 2015), to quantify the role of mixing state in determining CCN concentration (Ching et al., 2012, 2016, 2017) and aerosol optical properties (Fierce et al., 2016; Yao et al., 2022), and to determine structural uncertainty in more approximate aerosol models (Fierce et al., 2017; Zheng et al., 2021).

The methods for particle transport due to turbulent diffusion described in Curtis et al. (2017) and for mean wind advection described in this paper are based on the idea that the movement of particles between grid cells is represented by stochastic sampling. Importantly, the particle position within the grid cell is not tracked. This concept is therefore distinct from the particle-based Lagrangian techniques in the cloud modeling community (Heus et al., 2010; Arabas et al., 2015; Grabowski et al., 2018). These methods are similar to ours in that they also explicitly simulate microphysical processes on a population of computational particles (called super-particles in the cloud physics community), each representing a large number of real particles. However, they are different in that they simulate transport by tracking the super-droplet positions within the Eulerian grid.

There are advantages and disadvantages to each method. First, a stochastic algorithm can be constructed analogously to the finite volume transport schemes used in numerical weather models and chemical transport models, as we will show in this paper. This is beneficial for direct comparisons of different aerosol representations, which is one of our main motivations for developing particle-resolved aerosol models on the regional scale. Second, stochastic methods are more easily implemented in models that rely on different numerical grid structures, because they are based on the discretizations of the host model on the host grid. Lastly, stochastic methods for transport are computationally less expensive than tracking and updating particle positions throughout the simulation. However, stochastic transport algorithms have the disadvantage of numerical diffusion, similar to finite volume methods. This is in contrast to Lagrangian particle tracking methods that are inherently free of numerical diffusion.

The contribution of this paper is the development of stochastic transport algorithms for advection that remove the modeling limitations of the single-column model (WRF-PartMC-MOSAIC-SCM) by enabling a fully three-dimensional model to allow particle-resolved simulations on the regional scale (WRF-PartMC). WRF-PartMC is a tool for error quantification and benchmarking of traditional chemistry-transport models (e.g., WRF-Chem or CMAQ) that apply simplified aerosol mixing state representation, without the advection schemes being a potential source of differences.

The paper is structured as follows. Section 2 develops the stochastic particle advection method. Section 3 analyzes a series of four numerical experiments of increasing complexity, ranging from simple one-dimensional test cases with constant, uniform wind fields to a simulation with complex terrain and evolving meteorological fields. Section 4 summarizes our work. See Table G1 for a list of symbols used throughout the paper.

## 2 Stochastic particle transport scheme

This section describes the spatial and temporal discretization of the advection equation and then explains the stochastic sampling algorithm for the use in particle-resolved models. We will present the detailed derivation for one spatial dimension.

The generalization to three dimensions in space is straightforward and for brevity will not be explicitly written out, although see Sections 2.3 for notes on implementation details. In this study, we adopted the advection methods implemented by the host model WRF, rather than exploring alternative approaches. This choice ensures that future comparisons between WRF-PartMC and other aerosol representations in WRF-Chem will be fair and consistent. WRF-PartMC model was developed using WRFv3.9.1.1, coupled with chemistry for gas scalar transport and PartMC-MOSAIC for gas and aerosol chemistry with an additional interface for simulating stochastic particle transport.

## 2.1 Spatial and temporal discretization

The one-dimensional advection equation of a scalar quantity with (number) concentration $n(x,t)$ can be written as

$$\frac{\partial n(x,t)}{\partial t} = -u \frac{\partial n(x,t)}{\partial x}, \tag{1}$$

where $u > 0$ is the velocity of the advecting wind field (assumed to be constant in time and uniform in space here), $x$ is the spatial coordinate, and $t$ is time.

We discretize this equation spatially as

$$\frac{\partial n_i(t)}{\partial t} = -\frac{1}{\Delta x} \left( f_{i+\frac{1}{2}}(t) - f_{i-\frac{1}{2}}(t) \right), \tag{2}$$

where $\Delta x$ is the grid spacing in the $x$-coordinate, and $f_{i+\frac{1}{2}}(t)$ and $f_{i-\frac{1}{2}}(t)$ are the fluxes through the right and left grid cell boundaries of grid cell $i$ at time $t$, respectively, for $i = 0, \ldots, N_x - 1$.

The fluxes can be spatially discretized to different orders, with the WRF schemes of orders 1 to 6 written as (Wicker and Skamarock, 2002; Shu, 2009):

$$f_{i-\frac{1}{2}}^{1st} = un_{i-1}, \tag{3}$$

$$f_{i-\frac{1}{2}}^{2nd} = \frac{u}{2}(n_i + n_{i-1}), \tag{4}$$

$$f_{i-\frac{1}{2}}^{3rd} = \frac{u}{6}(2n_i + 5n_{i-1} - n_{i-2}), \tag{5}$$

$$f_{i-\frac{1}{2}}^{4th} = \frac{u}{12}(-n_{i+1} + 7n_i + 7n_{i-1} - n_{i-2}), \tag{6}$$

$$f_{i-\frac{1}{2}}^{5th} = \frac{u}{60}(-3n_{i+1} + 27n_i + 47n_{i-1} - 13n_{i-2} + 2n_{i-3}), \tag{7}$$

$$f_{i-\frac{1}{2}}^{6th} = \frac{u}{60}(n_{i+2} - 8n_{i+1} + 37n_i + 37n_{i-1} - 8n_{i-2} + n_{i-3}), \tag{8}$$

and similarly for the fluxes through the other boundary, $f_{i+\frac{1}{2}}$. We will explore the effect of using different orders of discretization in the context of stochastic particle-based advection in Sections 3.1 and 3.2.

For the temporal discretization, we use a 3rd-order Runge-Kutta method, analogous to the approach in WRF (Wicker and Skamarock, 2002), where the concentration at time $\ell + 1$ is calculated from the values at time $\ell$ as

$$n_i^{\dagger} = n_i^{\ell} - \frac{\Delta t}{3} \frac{1}{\Delta x}(f_{i+\frac{1}{2}}^{\ell} - f_{i-\frac{1}{2}}^{\ell}), \tag{9}$$

$$n_i^{\dagger\dagger} = n_i^{\ell} - \frac{\Delta t}{2} \frac{1}{\Delta x}(f_{i+\frac{1}{2}}^{\dagger} - f_{i-\frac{1}{2}}^{\dagger}), \tag{10}$$

$$n_i^{\ell+1} = n_i^{\ell} - \Delta t \frac{1}{\Delta x}(f_{i+\frac{1}{2}}^{\dagger\dagger} - f_{i-\frac{1}{2}}^{\dagger\dagger}), \tag{11}$$

where $\Delta t$ is the time step. The fluxes $f_{i+\frac{1}{2}}^{\dagger}$, $f_{i-\frac{1}{2}}^{\dagger}$, $f_{i+\frac{1}{2}}^{\dagger\dagger}$ and $f_{i-\frac{1}{2}}^{\dagger\dagger}$ are calculated as given in Eq. (3)–(8), using the concentrations $n_i^{\dagger}$ and $n_i^{\dagger\dagger}$, respectively.

Next, we will show how the discretized equations defined above are translated to specify the probabilities of particles moving between grid cells. Although we are presenting the method using the particular discretizations above, it is straightforward to derive stochastic versions of other spatial and temporal discretizations in the same way.

## 2.2 Stochastic sampling

To transform the method from Section 2.1 to a stochastic particle method, we consider a set of $N_i^{\ell}$ particles in grid cell $i$ at time step $\ell$. In reality, each particle will have an exact spatial location with a well-defined $x$ coordinate and will be moving with constant velocity $u$. However, in our stochastic method we will not track this per-particle spatial location and instead only track the set of particles in each grid cell. This is equivalent to the usual finite volume method of tracking the concentration in each grid cell, except that we are now sampling the concentration with a finite set of particles, allowing us to capture the high-dimensional variation in particle properties.

Note that the full WRF-PartMC model implementation explicitly tracks each particle in each grid cell so that it can store additional information about each particle (e.g., particle diameter, chemical constituents, etc.). In the following exposition we will not explicitly track the particles, but instead will only track the number of particles, $N_i^{\ell}$, in each grid cell. Section 2.4 contains further comments on translating the count-based scheme to a true per-particle method.

To advect the particles we relate the number of particles in each grid cell to the concentration in that grid cell, using

$$n_i^{\ell} = \frac{N_i^{\ell}}{V}, \tag{12}$$

where $V$ is the computational sampling volume within each grid cell. We can think of $V$ as controlling the "resolution" of the particle sampling and it will generally be much smaller than the true grid cell volume.

Having computed the values of $n_i^{\ell}$ for each grid cell, we then compute the finite volume fluxes $f_{i+\frac{1}{2}}^{\dagger\dagger}$ through each boundary from Eq. (11). This tell us that the average number of particles that should cross boundary $i + \frac{1}{2}$ is

$$\bar{F}_{i+\frac{1}{2}}^{\ell} = V \frac{\Delta t}{\Delta x} f_{i+\frac{1}{2}}^{\dagger\dagger}. \tag{13}$$

We interpret this probabilistically to mean that each of the $N_i^{\ell}$ particles has a probability of

$$p_{i+\frac{1}{2}}^{\ell} = \frac{\bar{F}_{i+\frac{1}{2}}^{\ell}}{N_i^{\ell}} \tag{14}$$

of crossing the boundary and leaving grid cell $i$. We then sample the number of particles that actually cross the boundary using a binomial distribution with $N_i^\ell$ trials and probability $p_{i+\frac{1}{2}}^\ell$, to give the discrete particle flux across the boundary to be

$$F_{i+\frac{1}{2}}^\ell = \text{Binom}\left(N_i^\ell, p_{i+\frac{1}{2}}^\ell\right). \tag{15}$$

Finally, we update the number of particles in each grid cell according to

$$N_i^{\ell+1} = N_i^\ell - F_{i+\frac{1}{2}}^\ell + F_{i-\frac{1}{2}}^\ell. \tag{16}$$

Note that this method obviously conserves the total number of discrete particles, because the $F_{i+\frac{1}{2}}^\ell$ particles that leave grid cell $i$ will all be transferred to grid cell $i+1$. In addition, because the mean of a binomial distribution is equal to the number of trials times the probability of success, we see that the average value of $N_i^1$ is exactly equal to $V n_i^1$ for the first time step. However, because the next time step will start from the stochastically sampled discrete value $N_i^1$, the average value of $N_i^2$ will not be exactly equal to the average value of $V n_i^2$.

Since probabilities larger than 1 are not meaningful, the time step needs to be chosen such that the probability (14) is less than or equal to 1. As a result, WRF-PartMC may need to take somewhat smaller time steps than required by the finite-difference advection in WRF.

## 2.3 Three-dimensional advection

The above derivation is for a one-dimensional domain, but the extension to three dimensions is straightforward. In three dimensions, we have a set of $N_{i,j,k}^\ell$ particles in grid cell $(i,j,k)$ at time step $\ell$. The fluxes are then defined as $f_{i+\frac{1}{2},j,k}^\ell$ and $f_{i,j+\frac{1}{2},k}^\ell$ and $f_{i,j,k+\frac{1}{2}}^\ell$ for the fluxes through the three positive boundaries of grid cell $(i,j,k)$, respectively. The fluxes through the other boundaries are defined similarly. The fluxes are then computed from a 3D finite volume discretization, but with the concentrations $n_i$ replaced by the number of particles $N_{i,j,k}^\ell$. This then yields probabilities of particles crossing each boundary by extending Eq. (14) from $i$ to three dimensions $(i,j,k)$. However, we now have three different probabilities, one for each boundary, corresponding to the three different directions in which particles can move. The time step should be chosen so that the sum of these probabilities is at most 1. We then sample the number of particles that move in each direction using a multinomial distribution with $N_{i,j,k}^\ell$ trials and probabilities $p_{i+\frac{1}{2},j,k}^\ell$, $p_{i,j+\frac{1}{2},k}^\ell$, and $p_{i,j,k+\frac{1}{2}}^\ell$ for the three directions, respectively. See Curtis et al. (2017) for a detailed description of the multinomial sampling algorithm. Finally, the number of particles in each grid cell is updated by extending Eq. (16) from one dimension $(i)$ to three dimensions $(i,j,k)$.

## 2.4 Explicit tracking of individual particles

Much of the power of a particle-based aerosol model is the ability to track the chemical composition and potentially morphology of individual particles, as is done by the PartMC (Riemer et al., 2009) model, which explicitly tracks a set of particles $\Pi_{i,j,k}^\ell$ in grid cell $(i,j,k)$ at time step $\ell$. To apply the stochastic advection algorithm of Section 2.2 to such a case, the stochastic fluxes can be computed using the total number, $N_{i,j,k}^\ell$, of particles in each grid cell, to give $F_{i+\frac{1}{2},j,k}^\ell$, $F_{i+\frac{1}{2},j,k}^\ell$, and $F_{i+\frac{1}{2},j,k}^\ell$ as the number of particles that will cross each boundary. However, rather than simply updating the particle counts using the fluxes,

we uniformly randomly sample $F^{\ell}_{i+\frac{1}{2},j,k}$ particles from the set $\Pi^{\ell}_{i,j,k}$ to move across the boundary, and similarly for the other two directions. This approach is used in the WRF-PartMC model.

## 2.5 Monotonicity

It is advisable in WRF-Chem simulations to use monotonic, positive-definite advection schemes (Wang et al., 2009; Chapman et al., 2009). WRF advection schemes without limiters have the tendency to overshoot as well as locally produce unrealistically low values. This is particularly problematic for chemical variables that have strong gradients due to the heterogeneity of emissions. The host WRF model features only a fifth order scheme for monotonic limiters. However due to the high computational expense of WRF-PartMC and the required domain decomposition to adequately meet that expense, we implemented third-order advection with monotonic limiters in WRF. This implementation utilized the existing third-order positive-definite scheme in WRF and applied the same limiter as used in the fifth-order monotonic scheme term (Skamarock, 2006; Wang et al., 2009). We focused on third- and fifth-order advection schemes because they combine good accuracy with some numerical dissipation at high spatial frequencies to suppress stochastic oscillations, as we will see in Sections 3.1 and 3.2.

## 2.6 Mixing ratio versus concentration

Many 3D atmospheric models such as WRF track the aerosol mixing ratio $q$ (units of $\#/\mathrm{kg}$) rather than the number concentration $n$ (units $\#/\mathrm{m}^3$) because this removes the need to adjust the tracer for changes in air density. However, the stochastic advection algorithm described above is based on the number concentration. In WRF-PartMC we convert the aerosol number concentration to a mass mixing ratio via $q = n/\rho$, compute mixing ratio fluxes using WRF's finite volume discretization, convert these back to number-concentration fluxes by multiplying by $\rho$, and then sample the stochastic particle transport with (14)–(16).

## 2.7 Variable sampling volumes and grid cell sizes

In Section 2.2 we assumed that the sampling volume $V$ is a constant. However, in the WRF-PartMC model the sampling volume is allowed to vary in space and time. This is done by defining a set of $V^{\ell}_{i,j,k}$ volumes in each grid cell $(i,j,k)$ at time step $\ell$, and allows the "particle resolution" to be adaptive to increase the accuracy while minimizing the computational cost. In such simulations a target number of particles per grid cell, $N_{\mathrm{p}}$, is chosen to be a fixed value and the sampling volume is then adapted using a halving/doubling procedure to maintain the actual number of particles per grid cell close to $N_{\mathrm{p}}$ (Riemer et al., 2009).

As described in detail in Curtis et al. (2017), the variable sampling volumes mean that the number of particles that move out of a grid cell (the "particle loss") is no longer generally equal to the number of particles that move into the neighboring cell (the "particle gain"). Instead, if $F^{\ell}_{i+\frac{1}{2},j,k}$ particles move out of grid cell $(i,j,k)$, then the number of particles that move into grid cell $(i+1,j,k)$ is scaled by the ratio of the sampling volumes. That is, the number of particles that move into grid cell

$(i+1, j, k)$ is

$$F^{\ell}_{i+\frac{1}{2}, j, k} \frac{V^{\ell}_{i+1, j, k}}{V^{\ell}_{i, j, k}}. \tag{17}$$

Similarly, if the grid cells have different physical volumes $\mathrm{Vol}^{\ell}_{i, j, k}$ then the above expression must be additional scaled by the ratio $\mathrm{Vol}^{\ell}_{i, j, k} / \mathrm{Vol}^{\ell}_{i+1, j, k}$. The advection algorithm in WRF-PartMC implements this scaling following the method in Curtis et al. (2017), which uses a variance-minimizing sampling algorithm that first samples the larger of the particle loss and gain terms and then subsamples from this to determine the other term. A potential concern is that the repeated resampling due to varying computational volumes, grid cell volumes, and air densities may cause the high-dimensional infomation carried by particles (see Section 2.4) to degenerate into overly similar representations. For example, if the particles carry a diameter sampled from a size distribution, the repeated resampling may cause the particles to converge to a single diameter. In Sec. 3.4 we investigate this numerically and see that it is not a significant issue in practice.

## 2.8 Computational cost

Regarding the computational costs of the finite volume, stochastic sampling, and Lagrangian particle tracking approaches, we consider a domain consisting of $N_{\mathrm{g}}$ grid cells and $N_{\mathrm{p}}$ computational particles per grid cell. The finite volume method, which only depends on the number of grid cells, has a cost $\mathcal{O}(N_{\mathrm{g}})$. In contrast, the Lagrangian particle tracking and stochastic methods depend on both number of grid cells and the number of particles. Therefore these methods scale as $\mathcal{O}(N_{\mathrm{g}} \times N_{\mathrm{p}})$ but the Lagrangian method has a higher cost as each particle must be checked and updated. In contrast, the cost of the stochastic method depends on the number of particles that actually move from one grid cell to another, which is frequently only a small fraction of the total number.

## 2.9 Comparison to Lagrangian particle tracking

With particles transported by deterministic advection there is no variance in the final position of particles that start in the same initial position. However, when we quantize space and only store which grid cell a particle is in, we can no longer move particles to the exact position where they should be located. That is, we are forced to incur some error. In a classic bias/variance tradeoff, we could achieve zero variance by moving all collocated particles to the same new grid cell, but this would result in an incorrect average position of the particles and a large bias. Alternatively, as we do in this paper, we can move some particles and not others, resulting in the correct mean velocity (zero bias) at the expense of introducing variance in particle position. Consequently, some particles will move faster and some slower than the mean velocity. To quantify the magnitude of this effect, see the example in Section 3.2 and Fig. 7.

## 3 Numerical experiments

The total error of a stochastic transport scheme can be bounded by two error terms that can be evaluated independently: (1) the stochastic error between the the stochastic solution and the finite volume solution, and (2) the deterministic error due to the

space-time discretization of the finite volume scheme. That is, for a stochastic solution $n^{\text{stoc}}$, a finite volume solution $n^{\text{FV}}$, and an exact true solution $n^{\text{true}}$, we can write:

$$\underbrace{\|n^{\text{stoc}} - n^{\text{true}}\|}_{\text{total error}} \leq \underbrace{\|n^{\text{stoc}} - n^{\text{FV}}\|}_{\text{stochastic error}} + \underbrace{\|n^{\text{FV}} - n^{\text{true}}\|}_{\text{deterministic error}}. \tag{18}$$

In this section we focus on the stochastic error. We do not consider the refinement of $\Delta x \to 0$ or $\Delta t \to 0$ as it is well understood how the finite volume methods converge to the true solution (deterministic error goes to zero) in these limits.

We present numerical examples of increasing complexity and discuss their convergence properties as the number of computational particles increases. Sec. 3.1 presents a one-dimensional test case with a soft-hat initial condition advected by a constant, uniform wind velocity to quantify the convergence as the number of computational particles increases. Sec. 3.2 simplifies the 1D test case to a uniform initial condition to study how the order of the advection scheme impacts convergence. Sec. 3.3, presents an idealized two-dimensional test case for solid-body rotational flow developed within WRF-PartMC using monotonic advection schemes. Finally, Sec. 3.4, shows stochastic particle transport for a realistic model domain and with realistic and evolving meteorological fields as simulated by the WRF-PartMC model.

To quantify the accuracy of the stochastic particle-resolved transport algorithm described above we use the relative root mean square error (RRMSE) between two solutions $n$ and $n'$ as given by

$$\text{RRMSE}(n, n') = \frac{\sqrt{\sum_i^{N_{\text{x}}} (n_i - n'_i)^2}}{\sqrt{\sum_i^{N_{\text{x}}} (n'_i)^2}}. \tag{19}$$

To determine the mean and confidence intervals for the RRMSE, we ran an ensemble of simulations with different random seeds for the stochastic sampling algorithm. The RRMSE was computed for each simulation run and then the overall mean and standard deviation were calculated, with the standard deviation being used to determine the 95% confidence interval for the RRMSE mean.

## 3.1 One-dimensional test case: Soft hat advected by uniform wind

We begin with the 1D soft-hat test case from Wicker and Skamarock (2002), with initial condition

$$n(x, t = 0) = 0.5 + \frac{0.5}{1 + \exp^{80(|x - 0.5| - 0.15)}}, \tag{20}$$

on the periodic domain $x \in [0, 1]$. Eq. 20 was modified from the expression in Wicker and Skamarock (2002) to include a background concentration so that there are some particles everywhere throughout the domain. The uniform velocity field was $u = 1$ (all quantities are taken as dimensionless here). For all presented results, the number of grid cells was $N_{\text{x}} = 50$ and the time step was $\Delta t = 0.008$, resulting in a Courant number of 0.4. The simulation duration was $T = 2$, giving two full revolutions of the domain. Simulation results were produced for first- through sixth-order advection schemes with no limiters applied.

Figure 1 shows the solution to the soft-hat problem after two revolutions ($t = 2$) for the finite volume method and one ensemble member of the stochastic method. Considering the finite volume solutions, we observe that the even-order methods

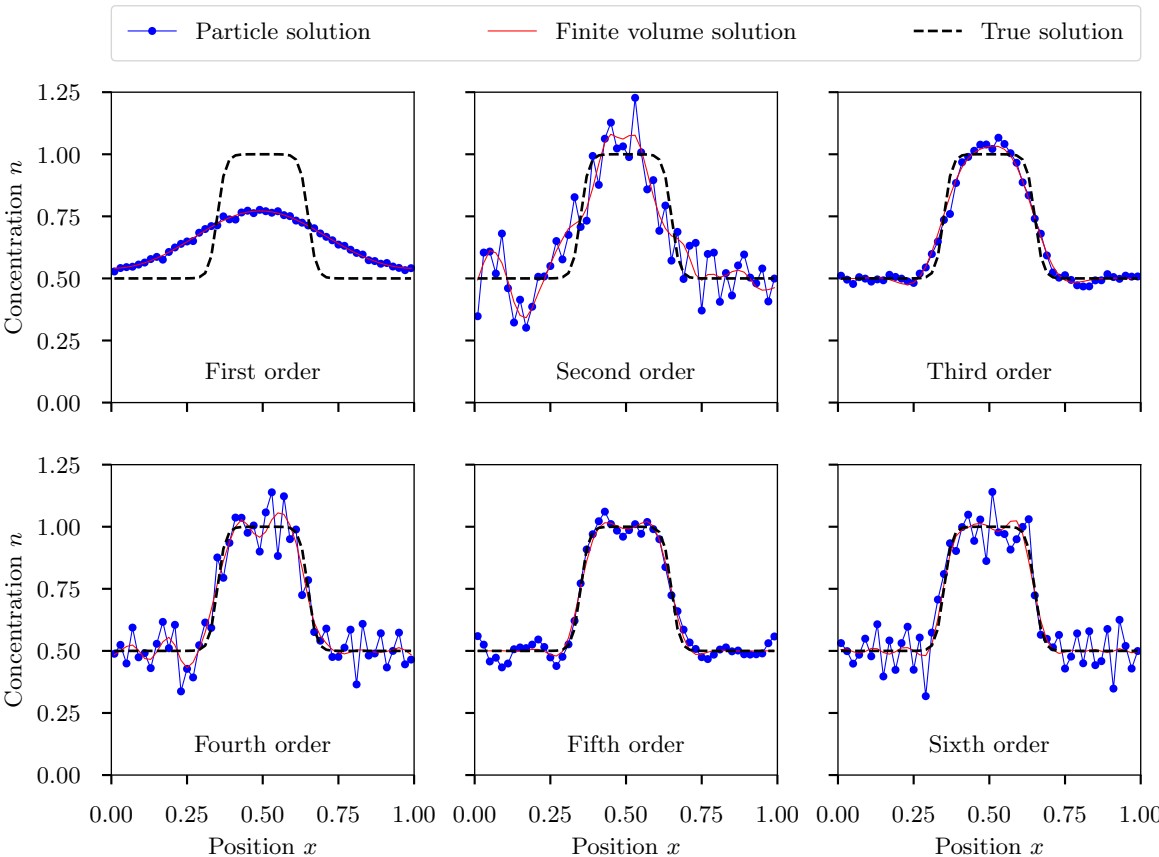

**Figure 1.** One dimensional soft-hat test case (Sec. 3.1): A single ensemble member of the stochastic solution is shown in blue for first to sixth-order methods, the deterministic finite volume solution is represented by the solid red line, and the analytical solution is shown as a black dashed line. The stochastic solution was simulated using $N_\mathrm{P} = 10^4$ computational particles per grid cell.

(2nd, 4th, 6th) produce more oscillatory solutions than the odd-order methods. This disparity between the even- and odd-order methods also occurs for the stochastic method, where the even-order solutions contain significantly more high-frequency noise than the odd-order solutions. We will analyze this phenomenon in more detail in Section 3.2.

We now turn to understanding the convergence of the stochastic particle method as the number of particles is increased. Figure 2(a) shows the ensemble mean error for the particle solution compared to the finite volume solutions for each order of advection, i.e., this is the error due to using a finite particle number but does not include any spatial discretization error as that is present for both the stochastic and finite volume methods. As the number of computational particles per grid cell, $N_\mathrm{p}$, increased, the solution converged to the deterministic finite volume solution. The rate of convergence for these stochastic methods is expected to be $\frac{1}{\sqrt{N_\mathrm{P}}}$ due to the central limit theorem and is denoted by the dashed line with slope $-\frac{1}{2}$. The stochastic error is largest for the even-order methods and smallest for the first-order method. In Sec. 3.2 we will show that this is because

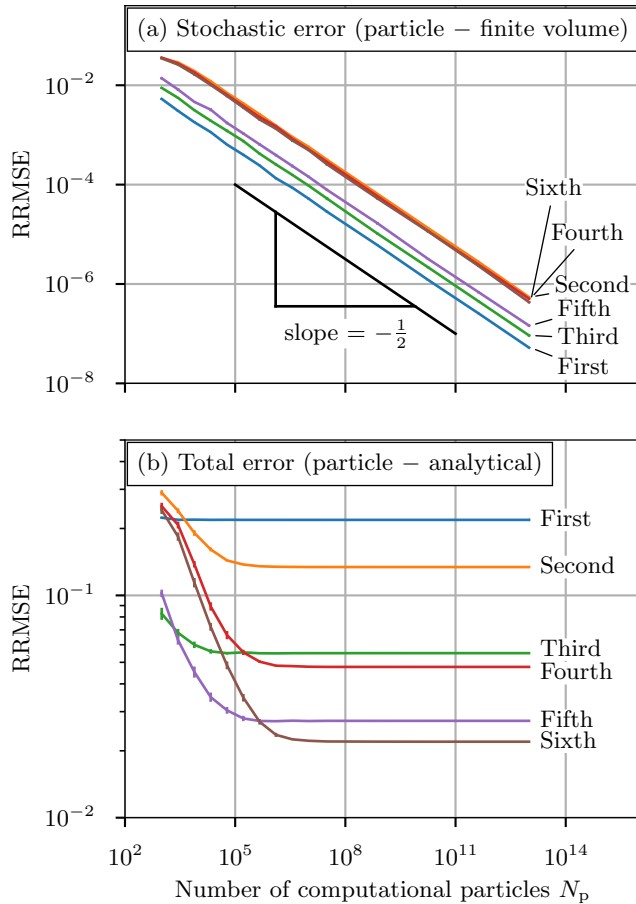

**Figure 2.** One dimensional soft-hat test case (Sec. 3.1): (a) Relative root-mean square error (RRMSE) between the stochastic particle solution and the deterministic finite volume solution for first- to sixth-order advection with varying number of computational particles. The dashed line shows the expected $\frac{1}{\sqrt{N_P}}$ convergence rate. (b) Relative root-mean square error (RRMSE) between the particle solution and the analytical solution for first- to sixth-order advection with varying number of computational particles per grid cell at $t = 2$. Error bars denote the 95% confidence interval as determined from an ensemble of 25 simulations.

the odd-order methods benefit from the damping of high frequency noise and the first-order method has the lowest stochastic error because it has the most damping.

Figure 2(b) shows the ensemble mean error for the particle solution compared to the analytical solution, i.e., the "total" error (finite volume error plus stochastic error). As the number of particles increases, the ensemble mean error approaches a constant value, which is the error due to the finite volume discretization. No matter how many computational particles are used, the total error cannot become smaller than the error introduced by the finite volume discretization. The finite volume error decreases in magnitude as the order of the advection method increases. For small values of $N_\mathrm{p}$, the stochastic error dominates the total error. The odd-order methods (third- and fifth-order) have lower stochastic error than the even-order methods which results in the total error converging to the finite volume error with fewer computational particles. In contrast, the first-order method has such large finite volume error, shown with the poor comparison of the finite volume solution to the true solution in Fig. 1, and such low stochastic error, shown in Fig. 2, that the finite volume error dominates the total error immediately. In all cases the finite volume error could also be reduced by decreasing the grid cell size, following the standard convergence analysis of finite volume methods (Durran, 2010).

In summary, these results show that the stochastic error of particle-resolved advection converges as expected with the rate of $\frac{1}{\sqrt{N_\mathrm{p}}}$. Conservative even-order schemes exhibit high-frequency oscillations in the finite-volume solution that are compounded by high-frequency noise from the stochastic sampling. For the dissipative odd-order schemes, numerical dissipation damps the high-frequency oscillations, as will be shown in Sec. 3.2. We therefore recommend the use of dissipative (odd-order) advection schemes. We also note that the stochastic advection scheme will be especially useful in open domains where we have an outflow boundary condition. In this case, the artificial noise injected by the stochastic sampling will be advected out of the domain and will not accumulate.

### 3.2   One-dimensional test case: Uniform concentration advected by uniform wind

The difference observed in Fig. 1 between the even- and odd-order solutions is of course due to the amount of numerical dissipation in the methods, where the even-order methods are conservative, while the odd-order methods have numerical dissipation of energy at high spatial frequencies (see, e.g., Durran (2010, §§3.3.2–3.3.3)). To understand how this interacts with the stochastic particle solution we derived a simple explicit model for the power spectrum of the stochastic solution. The details of this derivation are given in Appendices C–E. Briefly, we considered the uniform initial condition

$$n(x, t = 0) = 1 \tag{21}$$

on the periodic domain $x \in [0, 1]$ with the uniform velocity field $u = 1$. We used a computational volume of $V = 10\,000$ which thus means the stochastic particle system started with $N_\mathrm{p} = 10\,000$ particles per grid cell.

Figure 3 shows the solution to the case with uniform concentration and uniform wind after two revolutions for the finite volume method and for one ensemble member of the stochastic method using the same parameters as for Fig. 1, i.e., $N_\mathrm{x} = 50$ grid cells and a time step of $\Delta t = 0.008$ resulting in a Courant number of $0.4$. The exact solution to this problem is clearly $n(x, t) = 1$ for all $x$ and $t$, and the finite volume solution yields this solution exactly. However, as the stochastic method moves

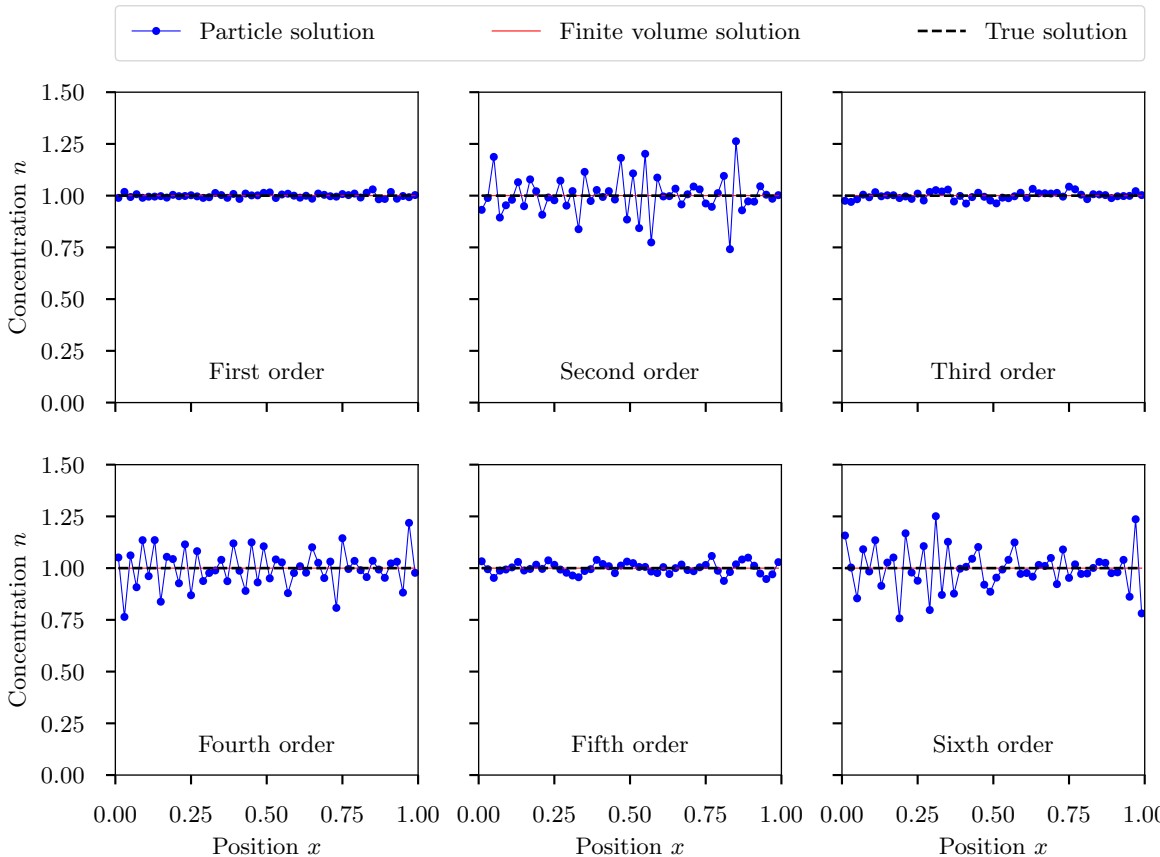

**Figure 3.** One dimensional uniform concentration advected by uniform wind: A single ensemble member of the stochastic solution is shown in blue for first to sixth-order methods, the determiniistic finite volume solution is represented by the solid red line and the analytical solution is shown as a black dashed line. The stochastic solution was simulated using $N_\mathrm{P} = 10^4$ computational particles per grid cell.

particles from one grid cell to the next, it is sampling a per-grid random number that is uncorrelated between grid cells, and is thus injecting energy at high spatial frequencies. As we will show with further analysis, this energy may then be dissipated by the numerical dissipation of the spatial discretization, and the question is whether the dissipation can effectively dampen the energy injection. To answer this question we will derive a simple model for the power spectrum of the stochastic solution.

We start by writing $\hat{n}_k$ for the discrete Fourier transform (DFT) of $n_i$, and recalling the classical fact that the power spectrum of the spatially-discretized system evolves according to

$$P_k^{\ell+1} = \exp(A_k)P_k^\ell, \tag{22}$$

where $P_k^\ell = |\hat{n}_k^\ell|^2$ is the power at wavenumber $k$ and time step $\ell$, and $A_k$ is the amplification factor at wavenumber $k$ (see Appendix A for details). Figure 4 shows the amplification factors for the different spatial discretizations. From this, we see

that the even-order methods have an amplification factor of zero at all wavenumbers, meaning that these methods are exactly

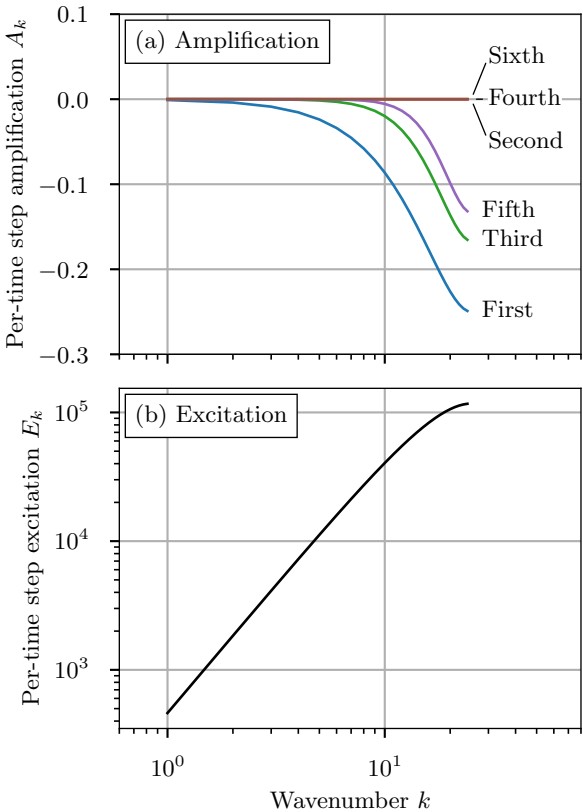

**Figure 4.** One dimensional uniform test case (Sec. 3.2): (a) Amplification factors, $A_k$, for the 1st to 6th order spatial discretizations. See Eq. (B8)–(B13) for details. (b) Excitation term, $E_k$. See Eq. (D12) for details.

conservative. In contrast, the odd-order methods have negative amplification factors at higher wavenumbers, showing that these methods will dissipate high spatial frequency components.

To understand the interaction between the stochastic sampling and the spatial discretization dissipation, Appendix D derives a recurrence relation for the power spectrum of an approximation to the stochastic solution:

$$\tilde{P}_k^{\ell+1} = \exp(A_k)\tilde{P}_k^\ell + E_k, \tag{23}$$

where $\tilde{P}_k^\ell$ is the power at wavenumber $k$ and time step $\ell$ of the approximate stochastic solution $\tilde{N}$, $A_k$ is the amplification factor of the spatial discretization, and $E_k$ is a stochastic excitation term (see Appendix D for details). Figure 4(b) plots the excitation term and we see that it is injecting energy at higher wavenumbers, due to the uncorrelated random noise in each grid cell from the stochastic transport. By comparing panels (a) and (b) in Fig. 4, we see that the negative amplification factors at

high wavenumbers will tend to suppress the energy injection.

To improve the clarity of results, we discretized with $N_x = 50$ grid cells, a time step of $\Delta t = 0.00125$, and total time $T = 2$ to give 1600 time steps for two revolutions and all simulations were run without limiters. Figure 5 shows the power spectrum of

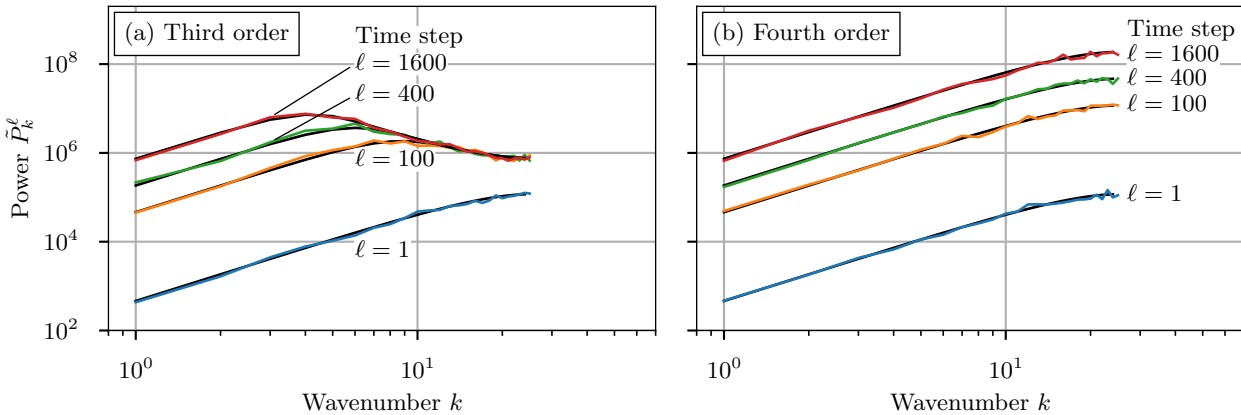

**Figure 5.** One dimensional uniform test case (Sec. 3.2): mean power spectra for third- and fourth-order advection methods after 1, 100, 400, and 1600 time steps. The overlaid black lines indicate the model prediction for each method at each time step. Each stochastic case was repeated 100 times to obtain the mean power spectra.

the stochastic solution for third- and fourth-order advection after 1, 100, 400 and 1600 time steps, with the black lines showing the model prediction for the power spectrum (see Appendix E for details). Here we see the constant energy injection at high
wavenumbers, with the fourth-order method steadily scaling up the power spectrum by the excitation term at each time step. In contrast, the third-order method has dissipation at higher wavenumbers which partially suppresses the injected energy and eventually reaches an equilibrium. This serves to suppress the high-frequency noise in the solution and explains the difference between the even- and odd-order stochastic solutions in Fig. 1.

     This point is further emphasized in Fig. 6 where all the stochastic methods are compared after 1600 time steps. The conser-
vative even-order schemes all fall on the same curve, increasing in power at higher frequencies. For the dissipative odd-order methods, high frequencies are damped. As expected from Fig. 4, the damping was least pronounced for the fifth-order method, and most pronounced for first order. In general, stochastic methods are less stable than their finite volume counterparts as the stochastic noise injects energy on average. Conservative even-order methods are unconditionally unstable due to this noise injection, because the scheme itself will never damp any of this additional energy.

Finally, to study the effect of spatial quantization where some particles move faster and some slower, causing variance in particle velocity and position (Sec. 2.9), let us consider the following example. If we assume a constant solution at all times (as in Appendix C), then the probability that a particle moves $k$ grid cells is $\mathrm{Binom}\left(k; N_\mathrm{t}, p\right)$, where $N_\mathrm{t}$ is the number of time steps and $p$ is the probability of moving each step, which will be equal to the Courant number. To investigate this, we refined the grid spacing and time step both by a factor of 10 to be $\Delta x = 0.002$ ($N_x = 500$) and $\Delta t = 0.0008$, which preserves the Courant
number of $C = p = 0.4$ of the original simulation, and we took $T = 1$ ($N_t = 1250$) for one revolution. Then, using the binomial distribution, the mean number of grid cells moved in one revolution is $\mu = N_t p = 500$, which is an exact approximation (zero bias), while the standard deviation is $\sigma = \sqrt{N_\mathrm{t} p(1-p)} = 17.3$. This corresponds to a physical distance of $x_\sigma = \sigma \Delta x = 0.035$.

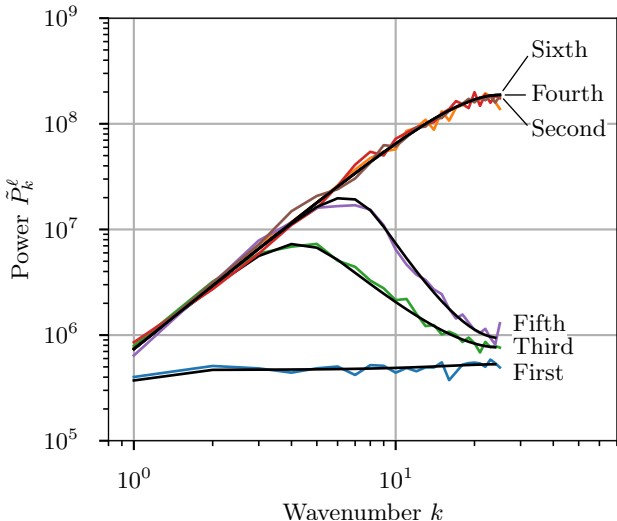

**Figure 6.** One dimensional uniform test case (Sec. 3.2): power spectra for first to sixth-order advection methods after 1600 time steps (two full revolutions of the system). The overlaid black lines indicate the model predictions for each method.

To understand the limiting behavior, we can use $N_t = T/\Delta t$ and $p = C = u\Delta t/\Delta x$ to rewrite $x_\sigma$ as

$$x_\sigma = \Delta x \sqrt{\frac{T}{\Delta t}\frac{u\Delta t}{\Delta x}(1-C)} = \sqrt{Tu(1-C)\Delta x}. \tag{24}$$

Now consider refining the grid ($\Delta x \to 0$) and time step ($\Delta t \to 0$) while keeping constant the Courant number C, the final time $T$ and the velocity $u$. In this limit, we can see that $x_\sigma \to 0$, so that the numerical diffusion of particles caused by the stochastic method vanishes.

Figure 7 shows the numerical result of the diffusion after one revolution for the particles originating in grid cell 250 (in blue), with the analytical binomial model shown in red. During sampling, some particles will travel faster and some will travel 360 slower, resulting in the binomial distribution of particles around the mean position.

### 3.3 Two-dimensional test case: Gaussian cone advected by solid body rotational wind field

To test the schemes in 2D, we used a scalar advection problem modified from Wicker and Skamarock (2002) where a Gaussian cone is advected in a square domain by a prescribed solid-body rotation flow. Simulations were conducted using third- and fifth-order monotonic advection schemes. Figure 8 shows the initial conditions. The domain is $100 \times 100$ nondimensional units 365 and the velocity field is defined as $u(x,y) = -\omega(y-50)$ and $v(x,y) = \omega(x-50)$ where $\omega = \frac{2\pi}{628}$. We took $\Delta x = \Delta y = 1$ and $\Delta t = 0.5$, so that one full rotation requires 1256 time steps. The maximum Courant number was 0.5. The initial particle mixing ratio was given as

$$q(x,y) = \max\left(10^{10}\exp\left(-\left(\frac{r}{r_0}\right)^2\right), 10^{-15}\right), \tag{25}$$

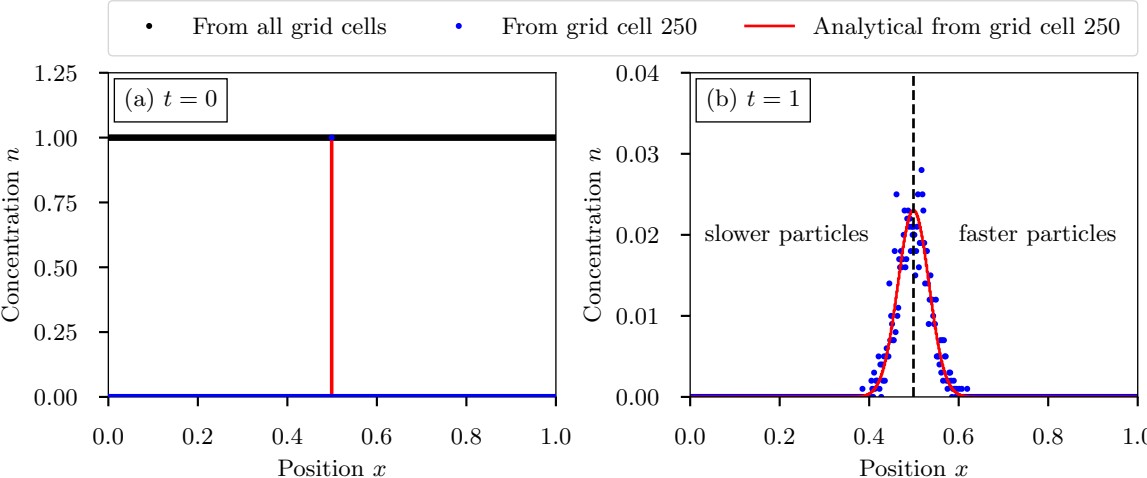

**Figure 7.** One dimensional uniform test case (Sec. 3.2) for the effect of spatial quantization on the stochastic solution: (a) Initial condition showing the uniform number concentration in all grid cells and the location of number concentration originating in grid cell 250 at about $x = 0.5$. (b) Number concentration of particles originating from grid cell 250 after one revolution at time $t = 1$ (blue points) with the analytical binomial model (solid red line). The vertical dashed line separates particles that moved too fast (to the right) and too slow (to the left).

where $r = \sqrt{(x - 50)^2 + (y - 75)^2}$ and $r_0 = 6$. The grid cell average values were constructed using $5 \times 5$-point Gaussian quadrature.

Figure 9 shows the solution after one revolution for the region of interest with 100, 1000 and 10 000 computational particles per grid cell as well as the finite volume solution. As the number of computational particles increased, the solution became less noisy and more similar to the finite-volume solution. This is quantified in Fig. 10, which shows the error of the particle solution compared to the finite volume solution. As expected, the stochastic error of the Monte Carlo method has a rate of convergence of $\frac{1}{\sqrt{N_P}}$. As the convergence of finite volume solutions to the analytical solution is well studied (Wicker and Skamarock, 2002), we do not include results showing convergence in $\Delta x$ and $\Delta y$.

### 3.4 Three-dimensional test case: Plume transported by WRF simulated meteorology

For this simulation we used WRF to fully simulate the meteorology, resulting in an evolving velocity field in 3D. We prescribed an idealized initial condition of particle mixing ratio and gas tracer mixing ratio for the model domain of Northern California. The gas tracer mixing ratio was used as a proxy for the solution of the finite volume method. The domain comprised $170 \times 160 \times 40$ grid cells, with $\Delta x = \Delta y = 4$ km and $\Delta z$ increasing logarithmically from an average value of 55 m near the surface to 650 m near the top of the model domain. The model time step was set to $\Delta t = 20$ s, ensuring that the sum of particle cell transfer probabilities did not exceed 1. For this case, an initial cloud of aerosol particle mixing ratio and gas mixing ratio was

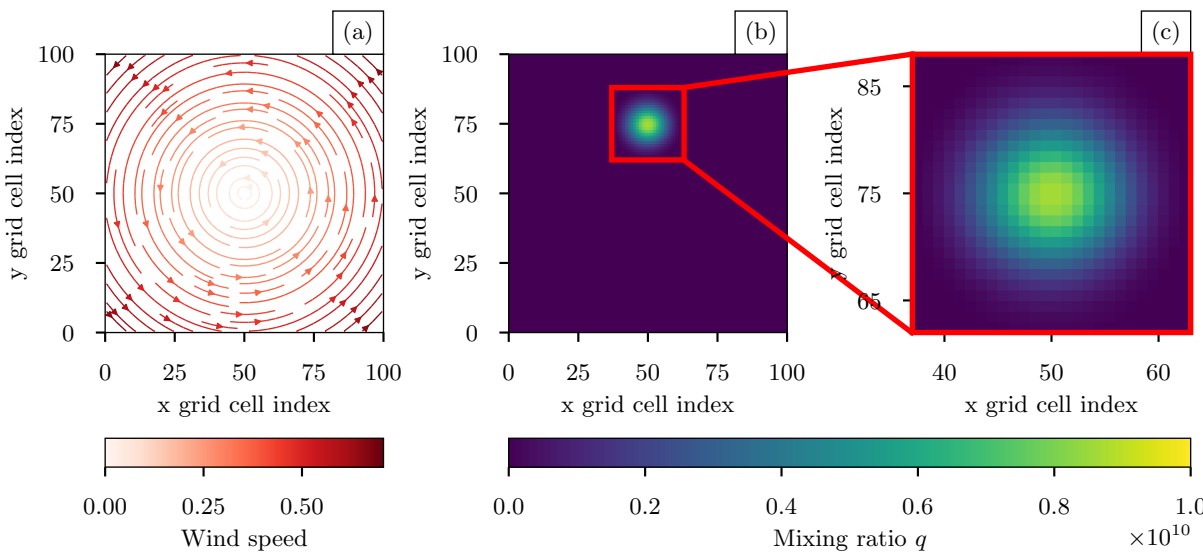

**Figure 8.** Two dimensional test case (Sec. 3.3): (a) rotational wind field, (b) the true solution of the Gaussian cone after one complete revolution, and (c) true solution of the red outlined region in (b).

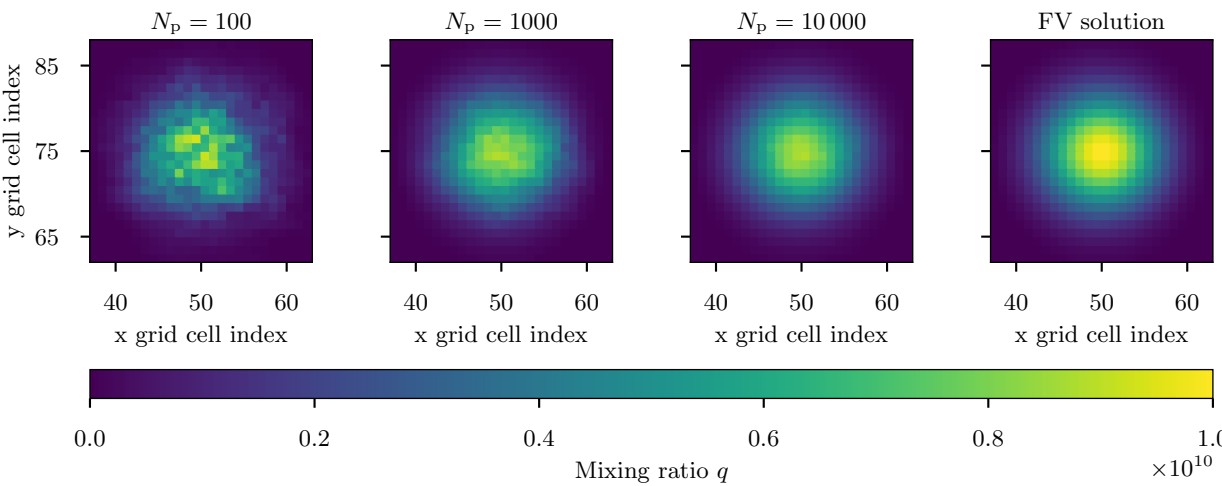

**Figure 9.** Two dimensional test case (Sec. 3.3): stochastic particle solution for 100, 1000 and 10 000 computational particles per grid cell and the finite volume solution after one revolution for the region shown in red in Fig. 8(b).

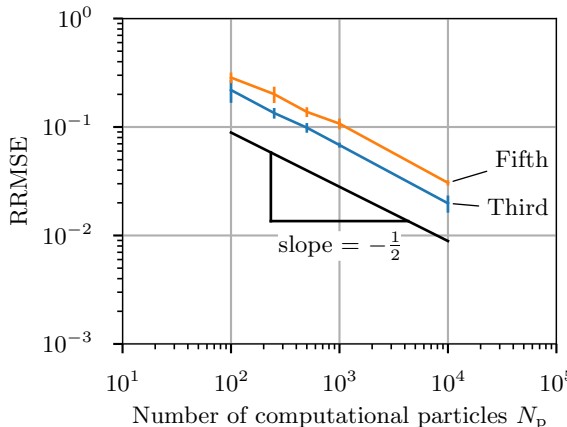

**Figure 10.** Two dimensional test case (Sec. 3.3): relative root-mean square error (RRMSE) between the particle solution and the finite volume solution for third- and fifth-order monotonic advection. Error bars indicate the 95% confidence interval from 10 simulations. The black reference line indicates the theoretical convergence rate with slope $\frac{1}{\sqrt{N_{\mathrm{p}}}}$.

determined by

$$
\quad q_{\mathrm{grid}}(x,y,z) = \max\left(10^{10}\exp\left(-\left(\sqrt{\left(\frac{x-x_0}{r_x}\right)^2 + \left(\frac{y-y_0}{r_y}\right)^2 + \left(\frac{z-z_0}{r_z}\right)^2}\right)\right), 10^{-15}\right), \tag{26}
$$

where $r_x = r_y = 6$ and $r_z = 4$, and the cloud is centered at grid cell $x_0 = 75$, $y_0 = 75$, and $z_0 = 1$. Here $q_{\mathrm{grid}}(x,y,z)$ is specified in grid coordinates (each grid cell is square of size $1 \times 1 \times 1$ grid units) before being transformed to physical coordinates for the simulation. Figure 11(a) shows the initial condition described by Eq. (26) at the lowest model layer. The initial condition was advected by the dynamic meteorology over a 12 hour period beginning at 0 UTC on 7 June 2010 using a time step of

$\Delta t = 20$ s. Meteorological initial and boundary conditions were based on analyses from the National Center for Environmental Predictions North American Mesoscale (NAM) model. The temporal evolution of the wind field is shown in Fig. 11(b)–(d) in increments of 6 hours. Gases and particles are subject only to advection and do not experience turbulent diffusion or any removal processes. Gas and aerosol boundary conditions were prescribed from initial values given in Eq. (26). When flow enters the domain at a boundary grid cell, the prescribed value is applied. Conversely, when flow exits the domain, the boundary grid

cell assumes a zero gradient condition, consistent with the host model WRF. Simulations were conducted using third- and fifth-order monotonic advection.

Figure 12 shows the solution after 12 hours for a varying number of computational particles per grid cell, with the finite volume solution for comparison. The simulation with 10 particles per grid cell is noisy as expected, capturing only general features of the particle number mixing ratio. As the number of computational particles was increased, the particle number

mixing ratio field became smoother and similar to the finite volume solution.

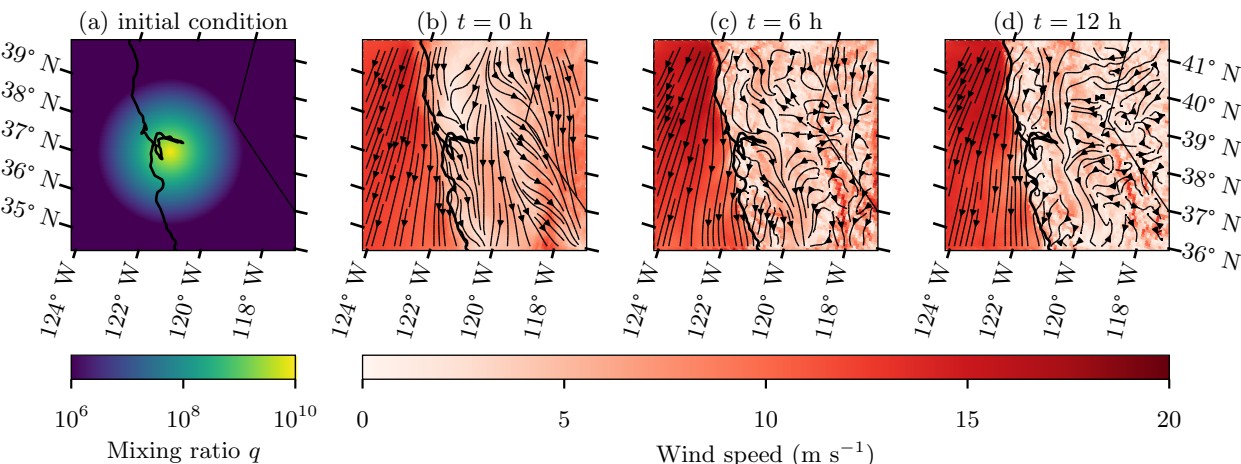

**Figure 11.** Three-dimensional test case (Sec. 3.4): (a) the initial condition and (b)-(d) snapshots of the wind velocity field at times $t = 0, 6$ and 12 h in the lowest model layer.

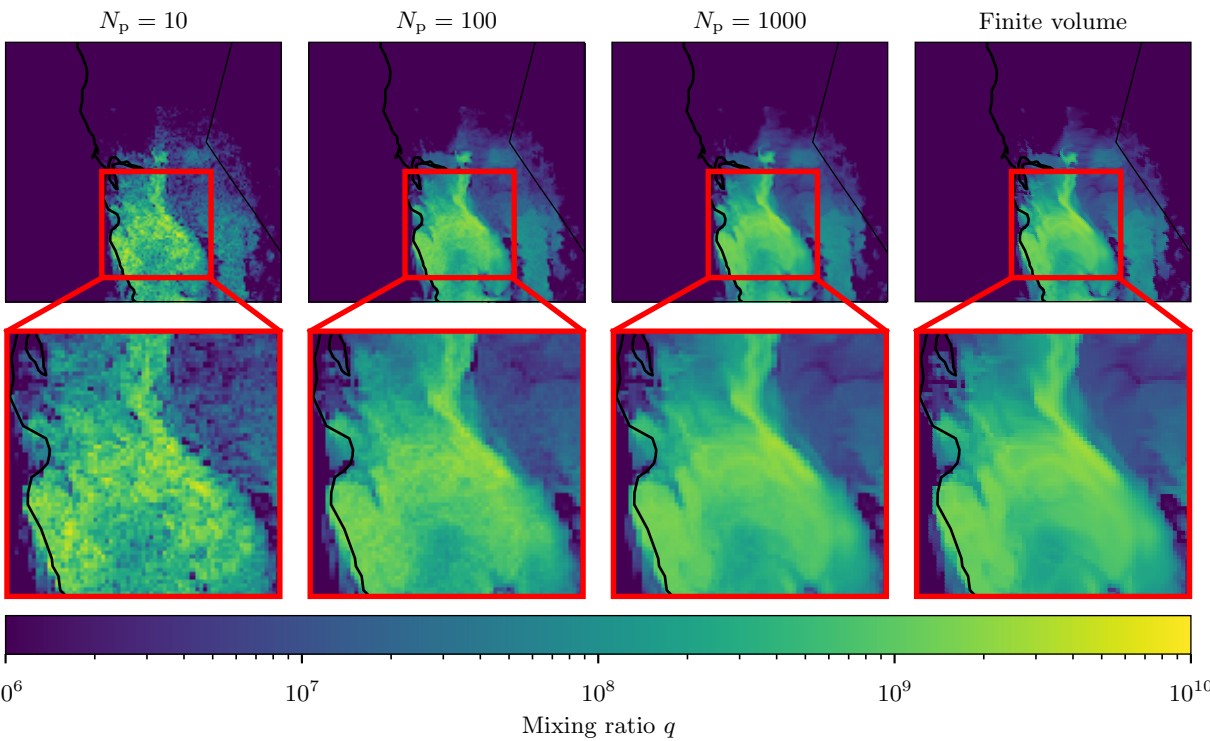

**Figure 12.** Three-dimensional test case (Sec. 3.4): lowest layer mixing ratios after 12 hours of simulation for 10, 100 and 1000 computational particles per grid cell, and the deterministic finite volume solution reference solution.

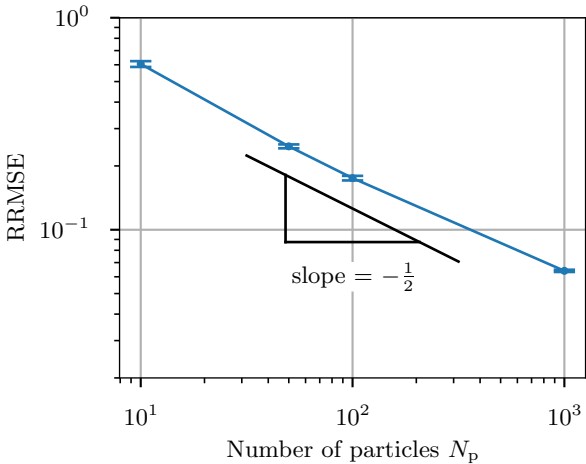

**Figure 13.** Three-dimensional test case (Sec. 3.4): convergence of the relative root-mean square error (RRMSE) between the stochastic solution and the finite volume solution as the number of computational particles per grid cell increases. Error bars show the 95% confidence interval from an ensemble of 5 simulations.

Figure 13 shows the convergence of the three-dimensional test case for third-order monotonic advection. As the number of computational particles increased, the error when compared to the finite volume solution converged at the expected rate of $\frac{1}{\sqrt{N_P}}$. Due to the stochastic nature of the problem, monotonic limiters may be applied to the number mixing ratio field that do not exist in the finite volume solution. As a result, a perfect $\frac{1}{\sqrt{N_P}}$ convergence rate is not expected.

Figure 14 confirms that the stochastic solution converges to the finite volume solution for the three-dimensional test case and that the variance decreases as the number of computational particles increases. For reference, Figure 14(a) shows an $x$-$y$ cross section of the mean mixing ratio in the lowest model layer at $t = 12$ h. The mean mixing ratio was calculated by averaging the stochastic solution over five simulations using $N_P = 100$ computational particles.

     Fig. 14(b)–(d) show different transects through the three-dimensional space and time. The star in Fig. 14(a) marks the location of the vertical mixing ratio profile (log-scaled) in Fig. 14(b) and the time series shown in Fig. 14(d). The red line denotes the transect shown in Fig. 14(c). The finite volume solution is compared to the ensemble mean of 10, 100 and 1000 computational particles with error bars denoting the 95% confidence interval. As the number of particles increased, the variance decreased and the solution converged to the finite volume solution.

     In Sec. 2.7, we discussed sampling complexities due to different computational volumes, grid cell volumes and air densities.
When these quantities substantially differ in adjacent grid cells, it could lead to undersampling of rare particle types. In our three-dimensional example, the largest ratio in density was 1.29, and the largest grid cell volume ratio was 1.96. For most of the grid cells, these ratios were closer to 1, indicated by domain average ratios of 1.01 and 1.11, respectively, at $t = 12$ h.

     To investigate whether undersampling occurred in practice, we ran the same scenario but sampled the particle diameter (a 1D attribute carried by particles) from a log-normal size distribution so that both rare large and small particles existed while

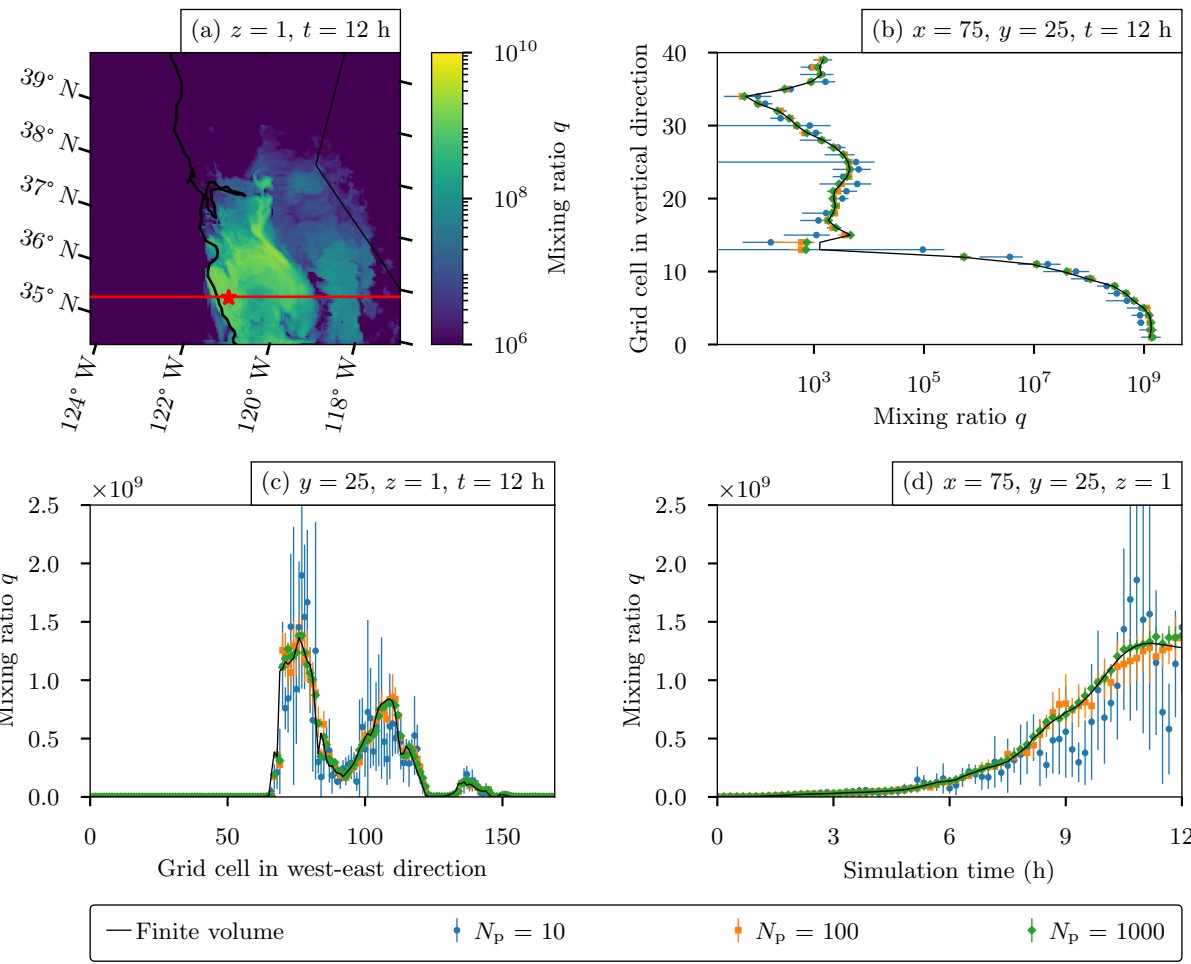

**Figure 14.** Three-dimensional test case (Sec. 3.4): (a) ensemble mean mixing ratio averaged over 5 simulations after 12 hours for the lowest model layer with $N_{\mathrm{p}} = 100$ computational particles per grid cell, (b) vertical profile of mixing ratio on logarithmic scale for stochastic solutions of $N_{\mathrm{p}} = 10$, 100 and 1000 computational particles per grid cell at $x = 75$, $y = 75$ at time $t = 12$h, (c) $x$ transect at $y = 25$ for stochastic solutions of $N_{\mathrm{p}} = 10$, 100 and 1000 computational particles per grid cell at $t = 12$h, and (d) time series at $x = 75$, $y = 25$ for $N_{\mathrm{p}} = 10$, 100 and 1000 computational particles per grid cell. The finite volume solutions for the profile, transect and time series are denoted by black lines. Points show means of 5 simultations and error bars denote the corresponding 95% confidence intervals.

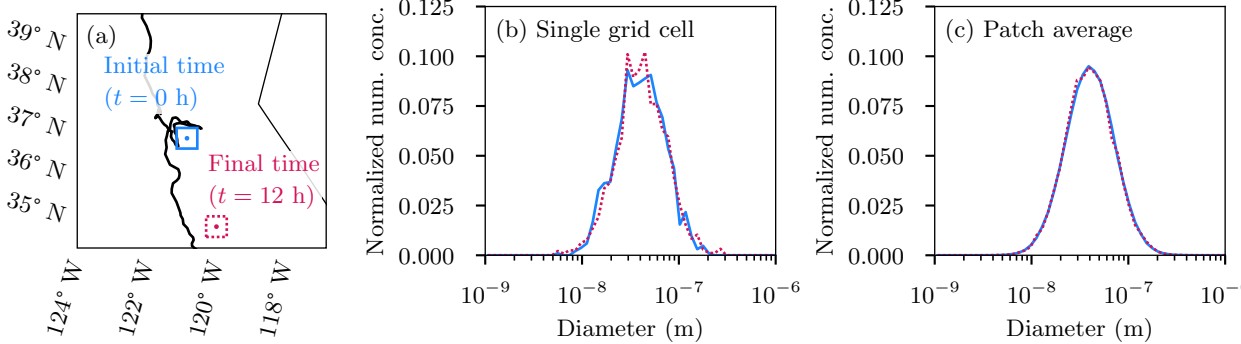

**Figure 15.** (a) Locations and patches where the normalized size distributions were computed. (b) Normalized size distributions at single grid cell locations shown in (a) using an ensemble of five simulations. (c) Mean normalized size distributions for the $15 \times 15$ subdomain patches shown in (a). Solid blue lines show the initial distributions computed at $t = 0$ h and at the initial point/region, while red dashed lines show the final distributions computed at $t = 12$ h and at the final point/region.

most computational particles resided in the center of the size distribution. We then compared the final size distributions with the initial size distributions to determine to what extent the rare large and small particles were systematically lost due to undersampling.

Figure 15(a) shows the locations for the initial and final size distribution plots. The locations of the initial and final points were chosen so that the final point is downwind of the initial point. All grid cells were initialized with 100 computational
particles drawn from a single log-normal mode, all with a constant geometric mean diameter and geometric standard deviation where only the magnitude of the distribution was adjusted. Figure 15(b) shows the normalized mean particle size distribution at the initial time and the final time at two single grid cells. Each distribution was averaged over five ensemble runs.

As we see from Fig. 15(b), the size distribution at the final time was similar to that at the initial time, with some stochastic noise. To reduce the stochastic noise, Fig. 15(c) shows the normalized mean particle size distribution at the initial time and the
final time for two $15 \times 15$ grid cell patches surrounding the points chosen for Fig. 15(b). Here the normalized size distributions were nearly identical, indicating that the size distribution information was not lost in the sampling procedure.

## 4    Conclusions

In this paper we presented the development of a stochastic particle advection method and demonstrated its performance for particle-resolved atmospheric aerosol transport in the combined WRF-PartMC model. The method is based on finite volume
advection schemes but interprets the fluxes as probabilities of particle transport, which can then be stochastically sampled. We analyzed the method in the one-dimensional setting to show that the stochastic particle sampling injects noise at high spatial frequencies and so the method performs best when using dissipative finite-volume discretizations, such as the third- and fifth-order schemes used in WRF.

We applied the new method in WRF-PartMC with the existing monotonic limiter for the fifth-order scheme and a new limiter for third order. We considered two test cases: a solid-body rotational wind field in 2D, and an atmospherically-relevant dynamic wind field over complex terrain in 3D. In both cases we observed the expected rates of convergence of the stochastic particle transport to the finite volume solution as the number of computational particles per grid cell was increased. For these examples, significant stochastic noise was evident in simulations with 100 computational particles per grid cell but stochastic noise was found to be less than 10% for simulations with 1000 particles per grid cell. This is considered a reasonable number of computational particles for large-scale WRF-PartMC simulations, as these simulations typically use on the order of 10 000 computational particles to accurately capture properties of the aerosol mixing state (Gasparik et al., 2020).

The value of this work is to enable direct comparison of particle-resolved aerosol representations to models that use approximate aerosol representations with simplified assumptions regarding size and composition (e.g., internally mixed modes or bins). Because the stochastic particle method is based on the same finite volume schemes used for the approximate representations, model comparisons can isolate the differences arising due to aerosol representation. Additionally, the new stochastic transport scheme allows the WRF-PartMC model to be used on the regional scale to quantify the impact of aerosol mixing state on climate-relevant aerosol properties, such as aerosol absorption and CCN concentration, and to compare these findings to existing studies (Matsui et al., 2013; Zhang et al., 2014; Zhu et al., 2016).

*Code and data availability.* WRF-PartMC version 1.0 is available at https://doi.org/10.5281/zenodo.10794890 (Curtis et al., 2024a). The current version of WRF-PartMC is available at https://github.com/open-atmos/wrf-partmc. The Python Jupyter notebooks and WRF-PartMC simulation data to reproduce figures contained within this manuscript are available at Curtis et al. (2024b).

*Author contributions.* JHC implemented the code and performed the data analysis. MW derived the analytical equations for the 1D model. All authors contributed to conceiving the numerical experiments and writing the manuscript.

*Competing interests.* The authors declare that they have no conflict of interest.

*Acknowledgements.* The authors are grateful to the two anonymous reviewers for their careful reading and constructive comments, which helped to significantly improve the manuscript. The authors acknowledge funding from ASR grants DOE DE-SC0019192 and DOE DE-SC0022130. This research is part of the Blue Waters sustained-petascale computing project, which is supported by the National Science Foundation (awards OCI-0725070 and ACI-1238993) and the state of Illinois. Blue Waters is a joint effort of the University of Illinois at Urbana-Champaign and its National Center for Supercomputing Applications. This work used Bridges-2 at Pittsburgh Supercomputing Center through allocation EES210036 from the Advanced Cyberinfrastructure Coordination Ecosystem: Services & Support (ACCESS) program, which is supported by National Science Foundation grants #2138259, #2138286, #2138307, #2137603, and #2138296.

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

## Appendix A: 1D advection in the frequency domain

To understand the behavior of the 1D deterministic and stochastic numerical methods it is helpful to write them in the frequency domain. To do this, we start in this section by considering only the deterministic (finite volume) case. We will then extend this to the stochastic case in the next section. We will use the vector notation

$$n = [n_0, n_1, \ldots, n_{N_\mathrm{x}-1}], \tag{A1}$$

$$f = [f_{\frac{1}{2}}, f_{3/2}, \ldots, f_{N_\mathrm{x}-\frac{1}{2}}]. \tag{A2}$$

We assume periodicity, so $n_i = n_{i+N_\mathrm{x}}$ and $f_{i-\frac{1}{2}} = f_{i-\frac{1}{2}+N_\mathrm{x}}$ for any $i$. Similarly, we encode the finite difference stencils as vectors:

$$r^{\mathrm{1st}} = [1, 0, \ldots, 0], \tag{A3}$$

$$r^{\mathrm{2nd}} = \frac{1}{2}[1, 0, \ldots, 0, 1], \tag{A4}$$

$$r^{\mathrm{3rd}} = \frac{1}{6}[5, -1, 0, \ldots, 0, 2], \tag{A5}$$

$$r^{\mathrm{4th}} = \frac{1}{12}[7, -1, 0, \ldots, 0, -1, 7], \tag{A6}$$

$$r^{\mathrm{5th}} = \frac{1}{60}[47, -13, 2, 0, \ldots, 0, -3, 27], \tag{A7}$$

$$r^{\mathrm{6th}} = \frac{1}{60}[37, -8, 1, 0, \ldots, 0, 1, -8, 37]. \tag{A8}$$

This allows us to express the fluxes (3)–(8) via a convolution:

$$f = ur * n, \tag{A9}$$

$$f_{i+\frac{1}{2}} = u \sum_{j=0}^{N_x-1} r_{i-j} n_j. \tag{A10}$$

Next, define the finite difference stencil

$$d = [1, -1, 0, \ldots, 0] \tag{A11}$$

so we can approximate the spatial derivative as

$$\frac{\partial n}{\partial x} \approx \frac{1}{\Delta x} d * n. \tag{A12}$$

Using this we can write the spatially discretized advection equation (2) as

$$\frac{\partial n}{\partial t} = -\frac{1}{\Delta x} d * f \tag{A13}$$

$$= -\frac{u}{\Delta x} d * r * n. \tag{A14}$$

We denote the discrete Fourier transform (DFT) using a hat, so $\hat{n} = \mathcal{F}(n)$ and similarly for other variables, and recall that the DFT is given by

$$\hat{n}_k = \sum_{j=0}^{N_x-1} n_j \exp(-i2\pi jk/N_x), \tag{A15}$$

where $i$ is the imaginary unit. Taking the DFT of (A14) gives

$$\frac{\partial \hat{n}_k}{\partial t} = -\frac{u}{\Delta x} d_k r_k n_k \tag{A16}$$

for each wavenumber $k$. The solution over one time step is then given by

$$\hat{n}_k^{\ell+1} = \exp(-C\hat{d}_k \hat{r}_k) \hat{n}_k^\ell, \tag{A17}$$

where $C$ is the Courant number given by

$$C = \frac{u\Delta t}{\Delta x}. \tag{A18}$$

Composing $\ell$ time steps gives the solution at time step $\ell$ as

$$\hat{n}_k^\ell = \exp(-\ell C \hat{d}_k \hat{r}_k) \hat{n}_k^0. \tag{A19}$$

To understand the numerical effect of the finite difference approximation we can compute the evolution of the power spectrum of the solution. The power spectrum is given by

$$P_k = |\hat{n}_k|^2 \tag{A20}$$

and the evolution of the power spectrum over one time step is given by

$$|\hat{n}_k^{\ell+1}|^2 = \hat{n}^{\ell+1}\hat{n}_k^{(\ell+1)*} \tag{A21}$$

$$= \left(\exp(-C\hat{d}_k\hat{r}_k)\hat{n}_k^\ell\right)\left(\exp(-C\hat{d}_k\hat{r}_k)\hat{n}_k^\ell\right)^* \tag{A22}$$

$$= \exp(-C\hat{d}_k\hat{r}_k)\exp(-C\hat{d}_k^*\hat{r}_k^*)\hat{n}_k^\ell\hat{n}_k^{\ell*} \tag{A23}$$

$$= \exp\left(-2C\operatorname{Re}(\hat{d}_k\hat{r}_k)\right)|\hat{n}_k^\ell|^2. \tag{A24}$$

The energy amplification of the method is thus given by

$$A_k = -2C\operatorname{Re}(\hat{d}_k\hat{r}_k) \tag{A25}$$

and we can write the power spectrum evolution as

$$P_k^{\ell+1} = \exp(A_k)P_k^\ell. \tag{A26}$$

If $A_k$ is zero then the method conserves the energy in wavenumber $k$, while negative values indicate that the method will dissipate energy with each time step.

## Appendix B: DFT of finite difference stencils

The DFT of the finite difference stencils $d$ and $r$ are found by applying (A15) to (A11) and (A3)–(A8). This gives

$$\hat{d}_k = 1 - \exp(-i2\pi k/N_{\mathrm{x}}) \tag{B1}$$

and

$$\hat{r}_k^{\mathrm{1st}} = 1, \tag{B2}$$

$$\hat{r}_k^{\mathrm{2nd}} = \frac{1}{2}\left(\exp(i2\pi k/N_{\mathrm{x}}) + 1\right), \tag{B3}$$

$$\hat{r}_k^{\mathrm{3rd}} = \frac{1}{6}\left(2\exp(i2\pi k/N_{\mathrm{x}}) + 5 - \exp(-i2\pi k/N_{\mathrm{x}})\right), \tag{B4}$$

$$\hat{r}_k^{\mathrm{4th}} = \frac{1}{12}\left(-\exp(i2\pi 2k/N_{\mathrm{x}}) + 7\exp(i2\pi k/N_{\mathrm{x}}) + 7 - \exp(-i2\pi k/N_{\mathrm{x}})\right), \tag{B5}$$

$$\hat{r}_k^{\mathrm{5th}} = \frac{1}{60}\left(-3\exp(i2\pi 2k/N_{\mathrm{x}}) + 27\exp(i2\pi k/N_{\mathrm{x}}) + 47 - 13\exp(-i2\pi k/N_{\mathrm{x}}) + 2\exp(-i2\pi 2k/N_{\mathrm{x}})\right), \tag{B6}$$

$$\hat{r}_k^{\mathrm{6th}} = \frac{1}{60}\left(\exp(i2\pi 3k/N_{\mathrm{x}}) - 8\exp(i2\pi 2k/N_{\mathrm{x}}) + 37\exp(i2\pi k/N_{\mathrm{x}}) + 37 - 8\exp(-i2\pi k/N_{\mathrm{x}}) + \exp(-i2\pi 2k/N_{\mathrm{x}})\right). \tag{B7}$$

The amplification $A_k$ of the above stencils can now be found by evaluating (A25) to give

$$A_k^{\text{1st}} = C\Big(-2 + 2\cos(2\pi k/N_{\text{x}})\Big) \tag{B8}$$

$$A_k^{\text{2nd}} = 0 \tag{B9}$$

$$A_k^{\text{3rd}} = \frac{C}{3}\Big(-3 + 4\cos(2\pi k/N_{\text{x}}) - \cos(2\pi 2k/N_{\text{x}})\Big) \tag{B10}$$

$$A_k^{\text{4th}} = 0 \tag{B11}$$

$$A_k^{\text{5th}} = \frac{C}{30}\Big(-20 + 30\cos(2\pi k/N_{\text{x}}) - 12\cos(2\pi 2k/N_{\text{x}}) + 2\cos(2\pi 3k/N_{\text{x}})\Big) \tag{B12}$$

$$A_k^{\text{6th}} = 0. \tag{B13}$$

## Appendix C: An approximate model for particle advection in 1D

We want to model the stochastic particle advection process as a deterministic advection process with some additional noise. We start by writing Equation (15) as

$$F_{i+\frac{1}{2}}^{\ell} = \text{Binom}\left(N_i^{\ell}, p_{i+\frac{1}{2}}^{\ell}\right) \tag{C1}$$

$$= \text{E}[F_{i+\frac{1}{2}}^{\ell}] + S_{i+\frac{1}{2}}^{\ell} \tag{C2}$$

$$= p_{i+\frac{1}{2}}^{\ell} N_i^{\ell} + S_{i+\frac{1}{2}}^{\ell} \tag{C3}$$

$$= \bar{F}_{i+\frac{1}{2}}^{\ell} + S_{i+\frac{1}{2}}^{\ell}, \tag{C4}$$

where $\bar{F}_{i+\frac{1}{2}}^{\ell}$ is the deterministic mean flux and $S_{i+\frac{1}{2}}^{\ell}$ is a zero-mean random variable representing the stochastic noise, given by

$$S_{i+\frac{1}{2}}^{\ell} = \text{Binom}\left(N_i^{\ell}, p_{i+\frac{1}{2}}^{\ell}\right) - \bar{F}_{i+\frac{1}{2}}^{\ell}. \tag{C5}$$

We approximate this stochastic noise by assuming that it is sampled from a constant uniform particle state with exactly $\check{N}$ particles per grid cell. From Equation (12) we have

$$\check{n} = \frac{\check{N}}{V} \tag{C6}$$

and because the velocity $u$ is constant and uniform the discretized flux is given by

$$\check{f}^{\dagger\dagger} = u\check{n}. \tag{C7}$$

From Equations (13) and (14) we then have

$$\breve{F} = V \frac{\Delta t}{\Delta x} \breve{f}^{\dagger\dagger} \tag{C8}$$

$$= V \frac{\Delta t}{\Delta x} u \frac{\breve{N}}{V} \tag{C9}$$

$$= C\breve{N}, \tag{C10}$$

$$\breve{p} = \frac{\breve{F}}{\breve{N}} \tag{C11}$$

$$= C. \tag{C12}$$

We can thus write the approximate stochastic noise by modifying Equation (C5) to give

$$\breve{S}_i = \mathrm{Binom}(\breve{N}, \breve{p}) - \breve{F} \tag{C13}$$

$$= \mathrm{Binom}(\breve{N}, C) - C\breve{N}. \tag{C14}$$

We want to write the approximate stochastic model in the frequency domain by taking a DFT. It is thus helpful to rewrite the equations in vector form, as we did in Section A. Similarly to Equations (A1) and (A2), we can write the particle counts $N_i^\ell$ and particle fluxes $F_{i-\frac{1}{2}}^\ell$ as vectors $N^\ell$ and $F^\ell$, and also do the same for other variables such as the average particle flux $\bar{F}_{i+\frac{1}{2}}^\ell$ and probabilities $p_{i+\frac{1}{2}}^\ell$.

Using the above vector notation and the difference stencil (A11) we can write the temporal update (16) as

$$N_i^{\ell+1} = N_i^\ell - F_{i+\frac{1}{2}}^\ell + F_{i-\frac{1}{2}}^\ell, \tag{C15}$$

$$N^{\ell+1} = N^\ell + d * F^\ell \tag{C16}$$

$$= N^\ell + d * \bar{F}^\ell + d * S^\ell. \tag{C17}$$

Taking the DFT now gives

$$\hat{N}_k^{\ell+1} = \hat{N}_k^\ell + \hat{d}_k \hat{F}_k^\ell + \hat{d}_k \hat{S}_k^\ell \tag{C18}$$

$$\approx \exp(-C\hat{d}_k \hat{r}_k)\hat{N}_k^\ell + \hat{d}_k \hat{S}_k^\ell \tag{C19}$$

$$\approx \exp(-C\hat{d}_k \hat{r}_k)\hat{N}_k^\ell + \hat{d}_k \hat{\breve{S}}_k. \tag{C20}$$

In Equation (C19) we approximated the update of the deterministic component with the exact solution of the deterministic advection equation, as in (A17). That is, we approximated the Runge-Kutta time step update with the exact solution. We then approximated the update of the stochastic component in (C20) by using the approximate stochastic noise $\breve{S}$.

Defining $\tilde{N}$ to be the solution of the approximate model, we can write the final approximate model from (C20) and (C14) as

$$\hat{\tilde{N}}_k^{\ell+1} = \exp(-C\hat{d}_k \hat{r}_k)\hat{\tilde{N}}_k^\ell + \hat{d}_k \hat{\breve{S}}_k, \tag{C21}$$

$$\breve{S}_i = \mathrm{Binom}(\breve{N}, C) - C\breve{N}. \tag{C22}$$

We observe that the approximate stochastic noise has mean and variance given by

$$\mathrm{E}[\breve{S}_i] = 0 \tag{C23}$$

$$\mathrm{Var}[\breve{S}_i] = C(1-C)\breve{N} \tag{C24}$$

for all $i$. The initial condition for the approximate model is given by $\tilde{N}_i^0 = \breve{N}$ for all $i$, which has DFT given by

$$\hat{\tilde{N}}_k^0 = N_{\mathrm{x}}\breve{N}\delta_{k,0}. \tag{C25}$$

## Appendix D: Recurrence relations for the first and second moments of the approximate model

Our aim is to solve the approximate model (C21) and (C22) analytically. Because the process is stochastic we will solve for the first two moments of the particle counts $\tilde{N}$ in the frequency domain and in this section we begin by deriving the appropriate recurrence relations.

Taking an expected value of (C21) gives the following recurrence relation for the first moment:

$$\mathrm{E}[\hat{\tilde{N}}_k^{\ell+1}] = \mathrm{E}\left[\exp(-C\hat{d}_k\hat{r}_k)\hat{\tilde{N}}_k^{\ell} + \hat{d}_k\hat{\breve{S}}_k\right] \tag{D1}$$

$$= \exp(-C\hat{d}_k\hat{r}_k)\mathrm{E}[\hat{\tilde{N}}_k^{\ell}] + \hat{d}_k\,\mathrm{E}[\hat{\breve{S}}_k] \tag{D2}$$

$$= \exp(-C\hat{d}_k\hat{r}_k)\mathrm{E}[\hat{\tilde{N}}_k^{\ell}] \tag{D3}$$

where we used the fact that the stochastic noise has zero mean.

Next we obtain a recurrence relation for the second moment of the particle counts. We use (C21) to compute

$$\mathrm{E}[\hat{\tilde{N}}_k^{\ell+1}\hat{\tilde{N}}_k^{(\ell+1)*}] = \mathrm{E}\left[\left(\exp(-C\hat{d}_k\hat{r}_k)\hat{\tilde{N}}_k^{\ell} + \hat{d}_k\hat{\breve{S}}_k\right)\left(\exp(-C\hat{d}_k\hat{r}_k)\hat{\tilde{N}}_k^{\ell} + \hat{d}_k\hat{\breve{S}}_k\right)^*\right] \tag{D4}$$

$$= \exp(-C\hat{d}_k\hat{r}_k - C\hat{d}_k^*\hat{r}_k^*)\mathrm{E}[\hat{\tilde{N}}_k^{\ell}\hat{\tilde{N}}_k^{\ell*}] + \hat{d}_k\exp(-C\hat{d}_k^*\hat{r}_k^*)\mathrm{E}[\hat{\tilde{N}}_k^{\ell}\hat{\breve{S}}_k^*] \tag{D5}$$

$$+ \exp(-C\hat{d}_k\hat{r}_k)\hat{d}_k\,\mathrm{E}[\hat{\breve{S}}_k\hat{\tilde{N}}_k^{\ell*}] + \hat{d}_k\hat{d}_k^*\,\mathrm{E}[\hat{\breve{S}}_k\hat{\breve{S}}_k^*] \tag{D6}$$

$$= \exp\left(-2C\,\mathrm{Re}(\hat{d}_k\hat{r}_k)\right)\mathrm{E}[\hat{\tilde{N}}_k^{\ell}\hat{\tilde{N}}_k^{\ell*}] + \hat{d}_k\hat{d}_k^*\,\mathrm{E}[\hat{\breve{S}}_k\hat{\breve{S}}_k^*]. \tag{D7}$$

In the final step above we used the fact that the approximate stochastic noise, $\hat{\breve{S}}_k$, has zero mean and is uncorrelated with the current solution, $\hat{\tilde{N}}_k$, because the noise is sampled from a fixed distribution, (C22), at each time step. This means that $\mathrm{E}[\hat{\tilde{N}}_k^{\ell}\hat{\breve{S}}_k^*] = \mathrm{E}[\hat{\breve{S}}_k\hat{\tilde{N}}_k^{\ell*}] = 0$ and so the cross terms vanish.

To compute the expected value of the squared magnitude of the stochastic noise, $\mathrm{E}[\hat{\breve{S}}_k\hat{\breve{S}}_k^*]$, we use (F7) and the statistics (C23) and (C24) of the stochastic noise to obtain

$$\mathrm{E}[\hat{\breve{S}}_k\hat{\breve{S}}_k^*] = N_{\mathrm{x}}^2\left|\mathrm{E}[\breve{S}_0]\right|^2\delta_{k,0} + N_{\mathrm{x}}\,\mathrm{Var}[\breve{S}_0] \tag{D8}$$

$$= N_{\mathrm{x}}C(1-C)\breve{N}. \tag{D9}$$

Substituting this into (D7) gives the recurrence relation

$$\mathrm{E}\left[|\hat{\tilde{N}}_k^{\ell+1}|^2\right] = \exp(A_k)\,\mathrm{E}\left[|\hat{\tilde{N}}_k^\ell|^2\right] + |\hat{d}_k|^2 N_{\mathrm{x}} C(1-C)\check{N}, \tag{D10}$$

where we have also used the amplification factor $A_k$ given by (A25).

Define the power at wavenumber $k$ by

$$\tilde{P}_k^\ell = \mathrm{E}\left[|\hat{\tilde{N}}_k^\ell|^2\right] \tag{D11}$$

and the excitation as

$$E_k = |\hat{d}_k|^2 N_{\mathrm{x}} C(1-C)\check{N}, \tag{D12}$$

where we can evaluate

$|\hat{d}_k|^2 = 2 - 2\cos(2\pi k/N_{\mathrm{x}}).$ \hfill (D13)

Using the above expressions we can write the final recurrence relation for the second moment (the power) as

$$\tilde{P}_k^{\ell+1} = \exp(A_k)\tilde{P}_k^\ell + E_k. \tag{D14}$$

The first term on the right-hand side represents the evolution of the second moment due to the discretized advection scheme, which may preserve the second moment or dissipate it depending on the scheme. This first term is identical to the evolution of
the power for the semi-discretization (A26). The second term on the right-hand side represents a constant injection of variance (energy) due to the stochastic noise.

## Appendix E: Analytical solution for the moments of the approximate model

In Appendix D we derived the recurrence relations for the first and second moments of the approximate model. In this section we solve these recurrence relations analytically. Starting with the first moment, the recurrence relation D3 has the solution

$$\begin{aligned}
\quad \mathrm{E}[\hat{\tilde{N}}_k^\ell] &= \exp(-\ell C \hat{d}_k \hat{r}_k)\,\mathrm{E}[\hat{\tilde{N}}_k^0] \tag{E1}\\
&= \exp(-\ell C \hat{d}_k \hat{r}_k) N_{\mathrm{x}} \check{N} \delta_{k,0} \tag{E2}\\
&= \exp(-\ell C \hat{d}_0 \hat{r}_0) N_{\mathrm{x}} \check{N} \delta_{k,0} \tag{E3}\\
&= N_{\mathrm{x}} \check{N} \delta_{k,0} \tag{E4}\\
&= \mathrm{E}[\hat{\tilde{N}}_k^0], \tag{E5}
\end{aligned}$$

where we used the initial condition (C25) and the fact that $\hat{d}_0 = 0$. From this we see that the first moment of the approximate model is constant in time and thus equal to its initial condition. We can write this as

$$\mathrm{E}[\tilde{N}_k^\ell] = \check{N}, \tag{E6}$$

for all $i$ and $\ell$. We thus see that the mean of the approximate model is identical to the solution (A19) of the deterministic spatial semi-discretization (A16), which is also constant for a uniform initial condition. That is, the approximate model mean is exactly the same as the exact time integration of the finite volume discretization, which is consistent with the observation that in Fig. 1 the particle solution oscillates around the finite volume solution.

To solve the recurrence relation for the second moment (D14) we first recall that the linear first-order recurrence relation

$$z^{\ell+1} = az^\ell + b \tag{E7}$$

for $a \in [0,1]$ has the solution

$$z^\ell = \begin{cases} a^\ell z^0 + (1-a^\ell)z^\infty & \text{if } a < 1, \\ z^0 + \ell b & \text{if } a = 1, \end{cases} \tag{E8}$$

where $z^0$ is the initial condition and $z^\infty$ is the steady state solution in the decaying case, given by

$$z^\infty = \frac{b}{1-a}. \tag{E9}$$

Applying this to (D14) gives

$$\tilde{P}_k^\ell = \begin{cases} \exp(\ell A_k)\tilde{P}_k^0 + \big(1 - \exp(\ell A_k)\big)\tilde{P}_k^\infty & \text{if } A_k < 0, \\ \tilde{P}_k^0 + \ell E_k & \text{if } A_k = 0, \end{cases} \tag{E10}$$

where the limiting moments are

$$\tilde{P}_k^0 = N_\mathrm{x}^2 \breve{N}^2 \delta_{k,0}, \tag{E11}$$

$$\tilde{P}_k^\infty = \frac{E_k}{1 - \exp(A_k)}, \tag{E12}$$

using (C25). To evaluate the above expression we need the amplification factors (B8)–(B13), the excitation (D12), and the squared magnitude (D13).

## Appendix F: Power identity for vectors with i.i.d. random components

Consider a vector of i.i.d. random variables $z_i$ for $i = 0, \ldots, N_\mathrm{x} - 1$. We want to compute the expected value of the squared magnitude of the DFT of this vector, i.e., $\mathrm{E}\big[|\hat{z}_k|^2\big]$ for each wavenumber $k$.

We start by observing that $\mathrm{E}[z_i z_j^*] = \mathrm{E}[z_i]\,\mathrm{E}[z_j] = \big|\mathrm{E}[z_0]\big|^2$ for $i \neq j$ because the random variables are independent. We also have $\mathrm{E}[z_i z_i^*] = \mathrm{E}\big[|z_i|^2\big] = \mathrm{E}\big[|z_0|^2\big] = \mathrm{Var}[z_0] + \big|\mathrm{E}[z_0]\big|^2$. We can thus write

$$\mathrm{E}[z_i z_j^*] = \big|\mathrm{E}[z_0]\big|^2 + \mathrm{Var}[z_0]\delta_{i,j}, \tag{F1}$$

for all $i, j$, where $\delta_{i,j}$ is the Kronecker delta.

We can now compute the expected value of the squared magnitude of the DFT:

$$\mathrm{E}\left[|\hat{z}_k|^2\right] = \mathrm{E}[\hat{z}_k\hat{z}_k^*] \tag{F2}$$

$$= \mathrm{E}\left[\left(\sum_{j=0}^{N_x-1} z_j\exp(-i2\pi jk/N_x)\right)\left(\sum_{\ell=0}^{N_x-1} z_\ell^*\exp(i2\pi\ell k/N_x)\right)\right] \tag{F3}$$

$$= \sum_{j=0}^{N_x-1}\sum_{\ell=0}^{N_x-1} \mathrm{E}[z_j z_\ell^*]\exp(-i2\pi(j-\ell)k/N_x) \tag{F4}$$

$$= \sum_{j=0}^{N_x-1}\sum_{\ell=0}^{N_x-1}\left(\left|\mathrm{E}[z_0]\right|^2 + \mathrm{Var}[z_0]\delta_{j,\ell}\right)\exp(-i2\pi(j-\ell)k/N_x) \tag{F5}$$

$$= \left|\mathrm{E}[z_0]\right|^2\sum_{j=0}^{N_x-1}\sum_{\ell=0}^{N_x-1}\exp(-i2\pi(j-\ell)k/N_x) + \mathrm{Var}[z_0]\sum_{j=0}^{N_x-1}\sum_{\ell=0}^{N_x-1}\delta_{j,\ell}\exp(-i2\pi(j-\ell)k/N_x). \tag{F6}$$

Consider the first term in the above expression. When $k = 0$ the sum is $N_x^2$ and when $k \neq 0$ the sum is zero because the inner sum consists of $N_x$ complex numbers that are spaced around the unit circle in a symmetric fashion. Now consider the second term. This collapses to $\sum_{j=0}^{N_x-1}\exp(-i2\pi(j-j)k/N_x) = N_x$ for all $k$. We thus have the final expression

$$\mathrm{E}\left[|\hat{z}_k|^2\right] = N_x^2\left|\mathrm{E}[z_0]\right|^2\delta_{k,0} + N_x\mathrm{Var}[z_0]. \tag{F7}$$

We see that the power spectrum consists of a uniform component that depends on the variance of the random variable and a DC component that depends on the mean of the random variable.

## Appendix G: Symbols used in this paper

Table G1 lists the symbols used in this paper.

**Table G1.** Symbols used in this paper.

| Symbol | Description | Reference |
|---|---|---|
| $A$ | Amplification factor | (A25) |
| $C$ | Courant number | (A18) |
| $d$ | Finite difference derivative stencil | (A11) |
| $\delta$ | Kronecker delta | (F1) |
| $\Delta t$ | Time step | (9)–(11) |
| $\Delta x$ | Spatial grid spacing | (2) |
| $E$ | Excitation | (D12) |
| $f$ | Concentration flux | (2) |
| $\bar{F}$ | Average particle flux | (13) |
| $\mathcal{F}$ | Discrete Fourier transform (DFT) | (A15) |
| $i$ | Spatial grid index | (2) |
| $k$ | Wavenumber index | (A15) |
| $\ell$ | Time step index | (9)–(11) |
| $n$ | Number concentration | (1) |
| $\hat{n}$ | DFT (discrete Fourier transform) of $n$ | (A15) |
| $N$ | Number of computational particles in a grid cell | §2.2, (12) |
| $\tilde{N}$ | Solution to the approximate model | (C21) |
| $\hat{\tilde{N}}$ | DFT (discrete Fourier transform) of $\tilde{N}$ | (C21) |
| $\check{N}$ | Initial particle number for the approximate model | (C6) |
| $N_\mathrm{p}$ | Number of computational particles per grid cell | (2) |
| $N_\mathrm{x}$ | Number of spatial grid points | (2) |
| $p$ | Probability | (14) |
| $P$ | Power spectrum of the semi-discrete solution | (A20) |
| $\tilde{P}$ | Power spectrum of the approximate model solution $\tilde{N}$ | (D11) |
| $\Pi$ | Particle set | §2.4 |
| $q$ | Mixing ratio | §2.6 |
| $r$ | Finite difference stencil coefficient | (A3)–(A8) |
| $S$ | Stochastic noise | (C5) |
| $t$ | Time | (1) |
| $T$ | Total simulation duration | §3 |
| $u$ | Velocity | (1) |
| $V$ | Computational volume | (12) |
| $x$ | Spatial coordinate | (1) |
| $y$ | Spatial coordinate | §3.3, §3.4 |
| $z$ | Spatial coordinate or generic complex variable | §3.4, §E, §F |