# Peer review of "Explicit stochastic advection algorithms for the regional scale particle-resolved atmospheric aerosol model WRF-PartMC (v1.0)"

_EGUsphere, 2024_

## Referee Comment (RC1)

**Review of "Explicit stochastic advection algorithms for the regional scale particle-resolved atmospheric aerosol model WRF-PartMC (v1.0)" by Jeffrey H. Curtis, Nicole Riemer, and Matthew West**

**Summary**

The paper presents a stochastic advection algorithm for particle-based models and its implementation in the particle-resolved atmospheric aerosol model WRF-PartMC. The algorithm is verified in a series of test cases of increasing complexity, starting from 1D advection in a periodic domain, and culminating in a three-dimensional simulation driven by realistic meteorology. Enabling regional simulations in WRF-PartMC is a significant model advancement. Therefore, the paper is suitable for publication in GMD after my minor comments and questions are addressed.

**General comments**

The proposed scheme is based on interpreting the fluxes coming from a numerical advection scheme as probabilities. This is valid only if the "probability" given by (14) is between 0 and 1. However, this may not always be the case. It seems to me that (14) will usually be very close numerically to the local Courant number. Some advection schemes are stable with Courant numbers greater than 1. This includes the WRF schemes used in this paper. Interestingly, in 1D the same condition that guarantees the probability to be less than 1 ($\frac{\Delta t}{\Delta x} f_{i+1/2} < n_i$) also guarantees positivity preservation. Yet, most advection schemes aren't positivity-preserving without additional limiting. Can the authors comment on this ? In the provided reproducibility notebook for the 1D test case there is code that clips the probability value, but there is no mention of this in the paper. Was this necessary and was a similar limiter used in the other test cases ?

The authors show that the stochastic transport algorithm injects energy at high spatial frequencies and analyze this process in considerable detail, including an approximate Fourier analysis. This analysis is very similar to von Neumann stability analysis of finite-difference schemes. Based on this the authors say that odd-order advection schemes are preferable, as they damp high spatial frequencies. I am wondering if a stronger statement could be made: that the stochastic algorithm based on even-order energy-conserving advection schemes is unconditionally unstable in a periodic domain, since it leads to unbounded growth of energy ? In general, are the stochastic algorithms less stable than their finite-volume base schemes ?

One of the motivations for using the proposed stochastic transport algorithm instead of Lagrangian advection is computational performance. Would it be possible to add to the article some performance numbers showing how much slower the stochastic algorithm is compared to its base finite-volume scheme ?

**Specific comments**

- Line 157 "However, we now have three different probabilities for each boundary, …" : If I understood the extension to three dimensions correctly, this sentence can be misleading. Maybe it would be better to say: "However, we now have three different probabilities, one for each boundary, …"
- Line 161: I think it is not (16), but a multi-dimensional extension of it, that needs to be used in three-dimensional simulations.
- Section 2.5: Please provide more information on the new monotonic third-order advection scheme. There are many approaches to constructing limiters for advection schemes. If the approach is something standard, like FCT, then I don't think it is necessary to provide every detail, but indicating which method was used and adding a citation would be helpful.
- Subsection 2.8 feels out of place to me in Section 2, since it details the computation of error metrics in numerical experiments. Maybe put it at the beginning of Section 3 ?
- In all numerical examples: Please indicate which experiments used the monotonic versions of the schemes and which the unlimited ones.
- Figure 2 caption: "at T = 2" should be "at t = 2"

- Lines 304-305 "The initial number concentration is given as …": In subsection 2.6 it is stated that `q` refers to the mixing ratio. Figures 6 and 7 are also labeled as mixing ratios. I realize that in this simple advection example the values are probably numerically equal. However, a similar issue is present in the subsequent realistic meteorology test case, where the text sometimes refers to the number concentration field, but, according to their labels, the figures are showing mixing ratios. For example, in line 328. It would be good if the language was consistent with the symbols and labels.
- Section 3.4 (WRF meteorology test case) : It would be helpful to provide more information about the setup of this test case. At which geographic location was the computational domain centered ? What was the time step ? What were the boundary conditions ?
- (E10): Shouldn't the conditions on the right be $A_k < 0$ and $A_k = 0$ since $a$ in (E7) corresponds to $\exp(A_k)$ ?

---

## Referee Comment (RC2)

**Review on article egusphere-2024-825**

This study proposes a novel approach for efficiently computing the transport of high-dimensional information of aerosol particles. It solves the evolution of particle number concentration using the finite volume method and calculates the transport of particles probabilistically based on the flux. This method has been thoroughly analyzed and tested in one-dimensional cases, and preliminary tests of two-dimensional and three-dimensional cases have also been presented. The paper is well-structured and easy to read. Despite the drawback of numerical diffusion affecting the mixing ratio of tracers, this approach is valuable for understanding the transport of aerosol particles. In addition, it could also help develop a Lagrangian particle tracking scheme in regional models. Therefore, it is suitable for publication in Geoscientific Model Development. I hope the comments provided below will contribute to the further improvement of the paper prior to its acceptance.

**General Comments**

I do not believe the flux-based approach is essentially different from particle-based Lagrangian tracking schemes. For example, a recent study has applied a quantization approach—similar to methods used in machine learning—to particle advection, effectively reducing computational costs and memory usage by quantizing particle positions within a cell (Matsushima et al., 2023). They round the positions of advected particles to the nearest possible locations within a cell, but it is also possible to use stochastic rounding instead of rounding to the nearest. Furthermore, the level of quantization can be optimized for the scientific objectives. Conversely, your method could also reconstruct Lagrangian particle trajectories probabilistically. I suggest investigating the impact of numerical diffusion more clearly by examining the variance in particle distribution, specifically how many particles move significantly faster or slower than the mean flow field. Such evaluations, particularly in two-dimensional test cases, could help clarify how your scheme differs from exact Lagrangian particle tracking.

The impact of resampling within your scheme requires further improvements. There are concerns that over time, the high-dimensional information held by particles may degenerate into overly similar states due to repeated resampling. Please consider adding a test case where 2–3 tracers are internally mixed to assess such effects on attribute-space dynamics. A simple yet effective analysis could compare the differences between the initial and final distributions in attribute space, providing insights into the extent of particle degeneracy. In addition, please provide details on variations of areas in each cell and atmospheric density and the results for the vertical cross sections of the tracer mixing ratio for the three-dimensional case to clarify the range of applicability of your scheme.

The dependency of the number of computational particles on the results is well presented in your scheme. However, convergence with respect to space and time resolution is not addressed. Additionally, it would be beneficial to specify the Courant number used in your test cases and discuss the limits of the Courant number that your scheme can accommodate.

**Specific Comments**

L152: The extension to multiple dimensions could be more clearly improved. In your method, unlike methods such as the corner transport upstream method or the conservative semi-Lagrangian method (Lin et al., 1996), it appears that you assume particles do not move in diagonal directions. Is my understanding correct? If so, would it be more reasonable to employ a flux that better aligns with the actual transport of particles?

L300: To check whether the schemes can be applied to simulate the mixing process, it would be better to adopt a swirling shear flow and a steeper initial condition like those of a cosine bell-type distribution and verify how well the tracer filament structures are preserved (see p.264 in Durran 2010).

**References**

- Matsushima, T., Nishizawa, S., and Shima, S.: Overcoming computational challenges to realize meter- to submeter-scale resolution in cloud simulations using the super-droplet method, Geosci. Model Dev., 16, 6211–6245, https://doi.org/10.5194/gmd-16-6211-2023, 2023.
- Lin, S.-J. and Rood, R. B.: Multidimensional flux-form semi-Lagrangian transport schemes, Mon. Weather Rev., 124, 2046–2070, https://doi.org/10.1175/1520-0493(1996)124<2046>2.0.CO;2, 1996.

- Durran, Dale R. Numerical methods for fluid dynamics: With applications to geophysics. Vol. 32. Springer Science & Business Media, 2010.

---

## Author Comment (AC1)

**Responses to Reviewer #1**

We thank the reviewer for taking the time to review our paper and for the constructive comments. The page and line numbers that we quote for indicating where we changed the manuscript refer to the revised marked-up version.

**(1.1)** The proposed scheme is based on interpreting the fluxes coming from a numerical advection scheme as probabilities. This is valid only if the "probability" given by (14) is between 0 and 1. However, this may not always be the case. It seems to me that (14) will usually be very close numerically to the local Courant number. Some advection schemes are stable with Courant numbers greater than 1. This includes the WRF schemes used in this paper. Interestingly, in 1D the same condition that guarantees the probability to be less than 1 ($\frac{\Delta t}{\Delta x} f_{i+1/2} < n_i$) also guarantees positivity preservation. Yet, most advection schemes aren't positivity-preserving without additional limiting. Can the authors comment on this ? In the provided reproducibility notebook for the 1D test case there is code that clips the probability value, but there is no mention of this in the paper. Was this necessary and was a similar limiter used in the other test cases ?

> Thank you for pointing out this issue. In the notebook example for the 1D test case, the clipping is indeed unnessessary since the Courant numbers are sufficiently low (we have chosen 0.4). We removed the corresponding lines of code from the notebook to avoid confusion.
>
> More generally, in the current implementation, we chose a time step that is small enough for the sum of the probabilities remaining smaller than or equal to 1 (i.e., we do not use any clipping). This requires WRF-PartMC to take somewhat smaller time steps than the finite-difference advection in WRF would require.
>
> We have made the following changes to the manuscript:
>
> - Line 160: "Since probabilities larger than 1 are not meaningful, the time step needs to be chosen such that the probability (14) is less than or equal to 1. As a result, WRF-PartMC may need to take somewhat smaller time steps than required by the finite-difference advection in WRF."
>
> - Line 170: "The time step should be chosen so that the sum of these probabilities is at most 1."
>
> - Line 395: "The model time step was set to $\Delta t = 20$ s, ensuring that the sum of the particle cell transfer probabilities does not exceed 1."

**(1.2)** The authors show that the stochastic transport algorithm injects energy at high spatial frequencies and analyze this process in considerable detail, including an approximate Fourier analysis. This analysis is very similar to von Neumann stability analysis of finite-difference schemes. Based on this the authors say that odd-order advection schemes are preferable, as they damp high spatial frequencies. I am wondering if a stronger statement could be made: that the stochastic algorithm based on even-order energy-conserving advection schemes is unconditionally unstable in a periodic domain, since it leads to unbounded growth of energy? In general, are the stochastic algorithms less stable than their finite-volume base schemes ?

> The reviewer is correct. The even-order energy-conserving is unconditionally unstable and in general, the energy injection will make stochastic methods less stable than the equivalent finite-volume based schemes. The following changes were made:
>
> - Line 355: "In general, stochastic methods are less stable than their finite volume counterparts as the stochastic noise injects energy on average. Conservative even-order methods are unconditionally unstable due to this noise injection, because the scheme itself will never damp any of this additional energy."

**(1.3)** One of the motivations for using the proposed stochastic transport algorithm instead of Lagrangian advection is computational performance. Would it be possible to add to the article some performance numbers showing how much slower the stochastic algorithm is compared to its base finite-volume scheme ?

We agree that there is benefit to adding some general discussion of the computational costs of the various methods, including finite volume, stochastic sampling and particle-tracking. At the same time, we feel that we placed too much emphasis on the computational cost advantages of our methods and too little emphasis on the main motivation for this paper, namely to create a stochastic transport algorithm that closely mimics the finite volume advection scheme (including its numerical diffusion). From our perspective, following the finite volume method closely is the main goal while the computational advantages of the stochastic sampling is more of a potentially helpful incidental benefit. To communicate this more clearly, we made the following changes:

- Beginning at Line 73, we made the following change to better emphasize the goals: "First, a stochastic algorithm can be constructed analogously to the finite volume transport schemes used in numerical weather models and chemical transport models, as we will show in this paper. This is beneficial for direct comparisons of different aerosol representations, which is one of our main motivations for developing particle-resolved aerosol models on the regional scale. Second, stochastic methods are more easily implemented in models that rely on different numerical grid structures, because they are based on the discretizations of the host model on the host grid. Lastly, stochastic methods for transport are computationally less expensive than tracking and updating particle positions throughout the simulation. However, stochastic transport algorithms have the disadvantage of numerical diffusion, similar to finite volume methods. This is in contrast to Lagrangian particle tracking methods that are inherently free of numerical diffusion."
- We added a new section 2.8, beginning at Line 224: "Regarding the computational costs of the finite volume, stochastic sampling, and Lagrangian particle tracking approaches, we consider a domain consisting of $N_\mathrm{g}$ grid cells and $N_\mathrm{p}$ computational particles per grid cell. The finite volume method, which only depends on the number of grid cells, has a cost $\mathcal{O}(N_\mathrm{g})$. In contrast, the Lagrangian particle tracking and stochastic methods depend on both number of grid cells and the number of particles. Therefore these methods scale as $\mathcal{O}(N_\mathrm{g} \times N_\mathrm{p})$ but the Lagrangian method has a higher cost as each particle must be checked and updated. In contrast, the cost of the stochastic method depends on the number of particles that actually move from one grid cell to another, which is frequently only a small fraction of the total number."

**(1.4)** Line 157 "However, we now have three different probabilities for each boundary, . . . " : If I understood the extension to three dimensions correctly, this sentence can be misleading. Maybe it would be better to say: "However, we now have three different probabilities, one for each boundary, . . . "

Thank you for pointing this out. We have improved the wording as suggested:

- Line 170: "However, we now have three different probabilities, one for each boundary, corresponding to the three different directions in which particles can move."

**(1.5)** Line 161: I think it is not (16), but a multi-dimensional extension of it, that needs to be used in three-dimensional simulations.

Thank you for catching this. It is indeed correct that Equation (16) must be extended to three dimensions by considering all possible fluxes. As a result, we made the following correction:

- Line 173: "Finally, the number of particles in each grid cell is updated by extending Eq. (16) from one dimension $(i)$ to three dimensions $(i, j, k)$."

**(1.6)** Section 2.5: Please provide more information on the new monotonic third-order advection scheme. There are many approaches to constructing limiters for advection schemes. If the approach is something standard, like FCT, then I don't think it is necessary to provide every detail, but indicating which method was used and adding a citation would be helpful.

> The third-order method is based directly off the limiter used for the fifth-order presented in Skamarock (2006) and further evaluated in Wang et al. (2009). This monotonic advection scheme uses a low-order monotonic flux limiter. We added the following to the manuscript:
>
> - Line 190: "This implementation utilized the existing third-order positive-definite scheme in WRF and applied the same limiter as used in the fifth-order monotonic scheme (Skamarock, 2006; Wang et al., 2009)."

**(1.7)** Subsection 2.8 feels out of place to me in Section 2, since it details the computation of error metrics in numerical experiments. Maybe put it at the beginning of Section 3 ?

> We followed the reviewer's suggestion and shifted the content of Section 2.8 to be at the end of the text that starts Section 3 (just before Section 3.1).

**(1.8)** In all numerical examples: Please indicate which experiments used the monotonic versions of the schemes and which the unlimited ones.

> We followed the reviewer's suggestion. The 1D Python notebook example contains no limiter while both WRF cases use monotonic limiters. We made the following additions to the manuscript to clearly denote this:
>
> - Line 276 (Section 3.1): "Simulation results were produced for first- through sixth-order advection schemes with no limiters applied."
> - Line 345 (Section 3.2): "...all simulations were run without limiters."
> - Line 376 (Section 3.3): "Simulations were conducted using third- and fifth-order monotonic advection schemes."
> - Line 408 (Section 3.4): "Simulations were conducted using third- and fifth-order monotonic advection."

**(1.9)** Figure 2 caption: "at $T = 2$" should be "at $t = 2$".

> The reviewer is correct that it should be time $t$ and not total time $T$. We have made the following correction:
>
> - In caption of Figure 2: "...the analytical solution for first- to sixth-order advection with varying number of computational particles per grid cell at $t = 2$. "

**(1.10)** Lines 304-305 "The initial number concentration is given as . . .": In subsection 2.6 it is stated that q refers to the mixing ratio. Figures 6 and 7 are also labeled as mixing ratios. I realize that in this simple advection example the values are probably numerically equal. However, a similar issue is present in the subsequent realistic meteorology test case, where the text sometimes refers to the number concentration field, but, according to their labels, the figures are showing mixing ratios. For example, in line 328. It would be good if the language was consistent with the symbols and labels.

The reviewer is correct that often number concentration and mixing ratio are numerically equal when density is uniform (such as in the 1D case and 2D case). However, we have made corrections throughout the manuscript in the cases where mixing ratio is truly the correct term to use.

**(1.11)** Section 3.4 (WRF meteorology test case) : It would be helpful to provide more information about the setup of this test case. At which geographic location was the computational domain centered ? What was the time step ? What were the boundary conditions ?

We made the following changes to the manuscript to more clearly define the simulation:

- Line 391: "We prescribed an idealized initial condition of particle mixing ratio and gas tracer mixing ratio for the model domain of Northern California."
- Line 393: "The domain comprised $170 \times 160 \times 40$ grid cells, with $\Delta x = \Delta y = 4$ km and $\Delta z$ increasing logarithmically from an average value of 55 m near the surface to 650 m near the top of the model domain."
- Line 402: "...beginning at 0 UTC on 7 June 2010 using a time step $\Delta t = 20$s."
- Line 403: "Meteorological initial and boundary conditions were based on analyses from the National Center for Environmental Predictions North American Mesoscale (NAM) model."
- Line 406: "Gas and aerosol boundary conditions were prescribed from initial values given in Eq. (26). When flow enters the domain at a boundary grid cell, the prescribed value is applied. Conversely, when flow exits the domain, the boundary grid cell assumes a zero gradient condition, consistent with the host model WRF."

**(1.12)** (E10): Shouldn't the conditions on the right be $A_k < 0$ and $A_k = 0$ since $a$ in (E7) corresponds to $\exp(A_k)$?

Thank you for catching this. We have made the correction to Equation (E10).

**References**

Skamarock, W. C.: Positive-definite and monotonic limiters for unrestricted-time-step transport schemes, Mon. Weather Rev., 134, 2241–2250, https://doi.org/10.1175/MWR3170.1, 2006.

Wang, H., Skamarock, W. C., and Feingold, G.: Evaluation of scalar advection schemes in the Advanced Research WRF model using large-eddy simulations of aerosol–cloud interactions, Mon. Weather Rev., 137, 2547–2558, https://doi.org/10.1175/2009MWR2820.1, 2009.

---

## Author Comment (AC2)

**Responses to Reviewer #2**

We thank the reviewer for taking the time to review our paper and for the constructive comments and suggestions. The page and line numbers that we quote for indicating where we changed the manuscript refer to the revised marked-up version.

**(2.1)** I do not believe the flux-based approach is essentially different from particle-based Lagrangian tracking schemes. For example, a recent study has applied a quantization approach—similar to methods used in machine learning—to particle advection, effectively reducing computational costs and memory usage by quantizing particle positions within a cell (Matsushima et al., 2023). They round the positions of advected particles to the nearest possible locations within a cell, but it is also possible to use stochastic rounding instead of rounding to the nearest. Furthermore, the level of quantization can be optimized for the scientific objectives. Conversely, your method could also reconstruct Lagrangian particle trajectories probabilistically. I suggest investigating the impact of numerical diffusion more clearly by examining the variance in particle distribution, specifically how many particles move significantly faster or slower than the mean flow field. Such evaluations, particularly in two-dimensional test cases, could help clarify how your scheme differs from exact Lagrangian particle tracking.

> Thank you for bringing this paper to our attention. We agree with the reviewer that when moving particles, one can either track exact positions or apply different levels of quantization of location. The choice to quantize results in a trade off of errors. The errors are either made in the mean speed, in the variance or some combination of mean speed and variance. The methods presented in our manuscript end up with the all the error in the variance. This error is equivalent to the numerical diffusion of the finite volume scheme. As a consequence of our choice, this means that some particles travel faster than mean speed while some travel slower, i.e., some particles will hop too many grid cells while some persist in a grid cell longer than they should.
>
> We have added the following to the manuscript to address this comment:
>
> - Line 238 (new Subsection 2.9 "Comparison to Lagrangian particle tracking"): "With particles transported by deterministic advection there is no variance in the final position of particles that start in the same initial position. However, when we quantize space and only store which grid cell a particle is in, we can no longer move particles to the exact position where they should be located. That is, we are forced to incur some error. In a classic bias/variance tradeoff, we could achieve zero variance by moving all collocated particles to the same new grid cell, but this would result in an incorrect average position of the particles and a large bias. Alternatively, as we do in this paper, we can move some particles and not others, resulting in the correct mean velocity (zero bias) at the expense of introducing variance in particle position. Consequently, some particles will move faster and some slower than the mean velocity. To quantify the magnitude of this effect, see the example in Section 3.2 and Fig. 7."
>
> - Line 358 (end of Section 3.2): "Finally, to study the effect of spatial quantization where some particles move faster and some slower, causing variance in particle velocity and position (Sec. 2.9), let us consider the following example. If we assume a constant solution at all times (as in Appendix C), then the probability that a particle moves $k$ grid cells is $\mathrm{Binom}\left(k; N_\mathrm{t}, p\right)$, where $N_\mathrm{t}$ is the number of time steps and $p$ is the probability of moving each step, which will be equal to the Courant number. To investigate this, we refined the grid spacing and time step both by a factor of 10 to be $\Delta x = 0.002$ ($N_x = 500$) and $\Delta t = 0.0008$, which preserves the Courant number of $C = p = 0.4$ of the original simulation, and we took $T = 1$ ($N_t = 1250$) for one revolution. Then, using the binomial distribution, the mean number of grid cells moved in one revolution is $\mu = N_t\, p = 500$, which is an exact approximation (zero bias), while the standard deviation is $\sigma = \sqrt{N_\mathrm{t}p(1 - p)} = 17.3$. This corresponds to a physical distance of $x_\sigma = \sigma \Delta x = 0.035$. To understand the limiting behavior, we can use

$N_t = T/\Delta t$ and $p = C = u\Delta t/\Delta x$ to rewrite $x_\sigma$ as

$$x_\sigma = \Delta x \sqrt{\frac{T}{\Delta t} \frac{u\Delta t}{\Delta x} (1 - C)} = \sqrt{Tu(1 - C)\Delta x}. \tag{1}$$

Now consider refining the grid ($\Delta x \to 0$) and time step ($\Delta t \to 0$) while keeping constant the Courant number C, the final time $T$ and the velocity $u$. In this limit, we can see that $x_\sigma \to 0$, so that the numerical diffusion of particles caused by the stochastic method vanishes.

Figure 7 shows the numerical result of the diffusion after one revolution for the particles originating in grid cell 250 (in blue), with the analytical binomial model shown in red. During sampling, some particles will travel faster and some will travel slower, resulting in the binomial distribution of particles around the mean position."

**(2.2)** The impact of resampling within your scheme requires further improvements. There are concerns that over time, the high-dimensional information held by particles may degenerate into overly similar states due to repeated resampling. Please consider adding a test case where 2–3 tracers are internally mixed to assess such effects on attribute-space dynamics. A simple yet effective analysis could compare the differences between the initial and final distributions in attribute space, providing insights into the extent of particle degeneracy. In addition, please provide details on variations of areas in each cell and atmospheric density and the results for the vertical cross sections of the tracer mixing ratio for the three-dimensional case to clarify the range of applicability of your scheme.

Thank you for these good suggestions. We added a new test case (Figure 15) to compare the initial and final distributions in attribute space (we used the 1D particle size as the attribute). We provided additional details on the cell variations and added a vertical profile to Figure 14 (used to be Figure 12). Specifically:

- Line 218: "A potential concern is that the repeated resampling due to varying computational volumes, grid cell volumes, and air densities may cause the high-dimensional infomation carried by particles (see Section 2.4) to degenerate into overly similar representations. For example, if the particles carry a diameter sampled from a size distribution, the repeated resampling may cause the particles to converge to a single diameter. In Sec. 3.4 we investigate this numerically and see that it is not a significant issue in practice."

- Page 24: Added new Figure 15.

- Line 429: "In Sec. 2.7, we discussed sampling complexities due to different computational volumes, grid cell volumes and air densities. When these quantities substantially differ in adjacent grid cells, it could lead to undersampling of rare particle types. In our three-dimensional example, the largest ratio in density was 1.29, and the largest grid cell volume ratio was 1.96. For most of the grid cells, these ratios were closer to 1, indicated by domain average ratios of 1.01 and 1.11, respectively, at $t = 12$ h. To investigate whether undersampling occurred in practice, we ran the same scenario but sampled the particle diameter (a 1D attribute carried by particles) from a log-normal size distribution so that both rare large and small particles existed while most computational particles resided in the center of the size distribution. We then compared the final size distributions with the initial size distributions to determine to what extent the rare large and small particles were systematically lost due to undersampling.

  Figure 15(a) shows the locations for the initial and final size distribution plots. The locations of the initial and final points were chosen so that the final point is downwind of the initial point. All grid cells were initialized with 100 computational particles drawn from a single log-normal mode, all with a constant geometric mean diameter and geometric standard deviation where only the magnitude of the distribution was adjusted. Figure 15(b) shows the normalized mean particle size distribution at the initial time and the final time at two single grid cells. Each distribution was averaged over five ensemble runs.

As we see from Fig. 15(b), the size distribution at the final time was similar to that at the initial time, with some stochastic noise. To reduce the stochastic noise, Fig. 15(c) shows the normalized mean particle size distribution at the initial time and the final time for two $15 \times 15$ grid cell patches surrounding the points chosen for Fig. 15(b). Here the normalized size distributions were nearly identical, indicating that the size distribution information was not lost in the sampling procedure."

- Added a new subfigure to Figure 14 that shows a vertical profile of the ensemble mean mixing ratio for 10, 100, 1000 computational particles.

- Line 424: "Fig. 14(b)–(d) show different transects through the three-dimensional space and time. The star in Fig. 14(a) marks the location of the vertical mixing ratio profile (log-scaled) in Fig. 14(b) and the time series shown in Fig. 14(d). The red line denotes the transect shown in Fig. 14(c). The finite volume solution is compared to the ensemble mean of 10, 100 and 1000 computational particles with error bars denoting the 95% confidence interval. As the number of particles increased, the variance decreased and the solution converged to the finite volume solution."

**(2.3)** The dependency of the number of computational particles on the results is well presented in your scheme. However, convergence with respect to space and time resolution is not addressed. Additionally, it would be beneficial to specify the Courant number used in your test cases and discuss the limits of the Courant number that your scheme can accommodate.

Thank you for raising this issue. The error in the stochastic solutions consists of two portions: (1) the stochastic error between the stochastic solution and the finite volume solution, which decreases with increasing number of computational particles, and (2) the error between the finite volume solution and the true solution, which decreases as the time step and grid spacing become smaller. The focus of this paper is on analyzing the convergence of the stochastic method to the finite volume method, while assuming that the standard finite volume methods presented here in WRF have the correct properties to converge as $\Delta x$ and $\Delta t$ are refined. We made changes to clarify this in the paper and added more details about the simulation:

- Line 248: "The total error of a stochastic transport scheme can be bounded by two error terms that can be evaluated independently: (1) the stochastic error between the the stochastic solution and the finite volume solution, and (2) the deterministic error due to the space-time discretization of the finite volume scheme. That is, for a stochastic solution $n^{\mathrm{stoc}}$, a finite volume solution $n^{\mathrm{FV}}$, and an exact true solution $n^{\mathrm{true}}$, we can write:

$$\underbrace{\|n^{\mathrm{stoc}} - n^{\mathrm{true}}\|}_{\text{total error}} \leq \underbrace{\|n^{\mathrm{stoc}} - n^{\mathrm{FV}}\|}_{\text{stochastic error}} + \underbrace{\|n^{\mathrm{FV}} - n^{\mathrm{true}}\|}_{\text{deterministic error}} ." \tag{2}$$

- Line 253: "We do not consider the refinement of $\Delta x \to 0$ or $\Delta t \to 0$ as it is well understood how the finite volume methods converge to the true solution (deterministic error goes to zero) in these limits."

- Line 275: "resulting in a Courant number of 0.4."

- Line 379: "The maximum Courant number was 0.5."

- Line 395: "The model time step was set to $\Delta t = 20$ s, ensuring that the sum of particle cell transfer probabilities did not exceed 1."

**(2.4)** L152: The extension to multiple dimensions could be more clearly improved. In your method, unlike methods such as the corner transport upstream method or the conservative semi-Lagrangian method (Lin et al., 1996), it appears that you assume particles do not move in diagonal directions. Is my understanding correct? If so, would it be more reasonable to employ a flux that better aligns with the actual transport of particles?

It is true that we do not consider particles moving in the diagonal directions. And it is correct that other finite volume schemes would result in an improved solution. However, the aim of this work was to develop methods that specifically mimic the advection schemes used in the WRF model. The reason for this is that it allows us to isolate the impact of aerosol representation when comparing WRF-PartMC to WRF-Chem.

In future work, the framework for simulating advection may be applied to other finite volume approaches. We made the following changes to clarify our choice:

- Line 55: "While we only present the development of stochastic advection schemes based on the finite volume methods in WRF, the methodology described here is applicable to any finite volume scheme or transport scheme such as Corner-Transport Upwind (Colella, 1990; LeVeque, 2002) or Flux-Form Semi-Lagrangian (Lin and Rood, 1996, 1997) that can be found in other host models."

- Line 128: "Although we are presenting the method using the particular discretizations above, it is straightforward to derive stochastic versions of other spatial and temporal discretizations in the same way."

**(2.5)** L300: To check whether the schemes can be applied to simulate the mixing process, it would be better to adopt a swirling shear flow and a steeper initial condition like those of a cosine bell-type distribution and verify how well the tracer filament structures are preserved (see p.264 in Durran 2010).

As described in the response to comment (2.3), our intention is to construct a stochastic method that mimics the existing finite volume methods in WRF. We are not aiming to improve the finite volume methods, but rather to understand the impact of the stochastic representation of aerosols on the model results. While test cases such as a swirling shear flow are interesting, our stochastic method will treat them in the same way as the finite volume method. Such examples are well-understood from the perspective of finite volume methods, and we thus feel that such examples are not necessary for this paper.

**References**

Colella, P.: Multidimensional upwind methods for hyperbolic conservation laws, J. Comput. Phys., 87, 171–200, 1990.

LeVeque, R. J.: Multidimensional Scalar Equations, p. 447–468, Cambridge Texts in Applied Mathematics, Cambridge University Press, 2002.

Lin, S.-J. and Rood, R. B.: Multidimensional flux-form semi-Lagrangian transport schemes, Mon. Weather Rev., 124, 2046–2070, 1996.

Lin, S.-J. and Rood, R. B.: An explicit flux-form semi-Lagrangian shallow-water model on the sphere, Q. J. R. Meteorolog. Soc., 123, 2477–2498, 1997.